# Evolution of Yin and Yang isoforms of a chromatin remodeling subunit precedes the creation of two genes

Wen Xu[1], Lijiang Long[1,2], Yuehui Zhao[1], Lewis Stevens[3], Irene Felipe[4], Javier Munoz[5], Ronald E Ellis[6], Patrick T McGrath[1,7,8]*

[1]School of Biological Sciences, Georgia Institute of Technology, Atlanta, United States; [2]Interdisciplinary Graduate Program in Quantitative Biosciences, Georgia Institute of Technology, Atlanta, United States; [3]Institute of Evolutionary Biology, Ashworth Laboratories, School of Biological Sciences, University of Edinburgh, Edinburgh, United Kingdom; [4]Epithelial Carcinogenesis Group, Spanish National Cancer Research Center-CNIO, Madrid, Spain; [5]Proteomics Unit-ProteoRed-ISCIII, Spanish National Cancer Research Center-CNIO, Madrid, Spain; [6]Department of Molecular Biology, Rowan University School of Osteopathic Medicine, Stratford, United States; [7]Parker H. Petit Institute of Bioengineering and Bioscience, Georgia Institute of Technology, Atlanta, United States; [8]School of Physics, Georgia Institute of Technology, Atlanta, United States

*For correspondence:
patrick.mcgrath@biology.gatech.edu

Competing interests: The authors declare that no competing interests exist.

**Abstract** Genes can encode multiple isoforms, broadening their functions and providing a molecular substrate to evolve phenotypic diversity. Evolution of isoform function is a potential route to adapt to new environments. Here we show that de novo, beneficial alleles in the *nurf-1* gene became fixed in two laboratory lineages of *C. elegans* after isolation from the wild in 1951, before methods of cryopreservation were developed. *nurf-1* encodes an ortholog of BPTF, a large (>300 kD) multidomain subunit of the NURF chromatin remodeling complex. Using CRISPR-Cas9 genome editing and transgenic rescue, we demonstrate that in *C. elegans*, *nurf-1* has split into two, largely non-overlapping isoforms (NURF-1.D and NURF-1.B, which we call Yin and Yang, respectively) that share only two of 26 exons. Both isoforms are essential for normal gametogenesis but have opposite effects on male/female gamete differentiation. Reproduction in hermaphrodites, which involves production of both sperm and oocytes, requires a balance of these opposing Yin and Yang isoforms. Transgenic rescue and genetic position of the fixed mutations suggest that different isoforms are modified in each laboratory strain. In a related clade of *Caenorhabditis* nematodes, the shared exons have duplicated, resulting in the split of the Yin and Yang isoforms into separate genes, each containing approximately 200 amino acids of duplicated sequence that has undergone accelerated protein evolution following the duplication. Associated with this duplication event is the loss of two additional *nurf-1* transcripts, including the long-form transcript and a newly identified, highly expressed transcript encoded by the duplicated exons. We propose these lost transcripts are non-functional side products necessary to transcribe the Yin and Yang transcripts in the same cells. Our work demonstrates how gene sharing, through the production of multiple isoforms, can precede the creation of new, independent genes.
DOI: https://doi.org/10.7554/eLife.48119.001

## Introduction

There is general interest in understanding how animals adapt to new environments. What are the alleles that matter to positive selection and what sort of genes do they target? Since methods were developed to map and identify the genes harboring causative genetic variation, researchers have often isolated changes in the same gene in different populations or species (*Wood et al., 2005*; *Martin and Orgogozo, 2013*). Besides targeting specific genes, evolution can target classes of genes that share molecular features such as biochemical (*e.g.* chemoreceptor genes; *Bachmanov and Beauchamp, 2007*; *Keller et al., 2007*; *Wisotsky et al., 2011*; *Lunde et al., 2012*; *McRae et al., 2012*; *McBride et al., 2014*; *Greene et al., 2016a*; *Greene et al., 2016b*) or developmental function (*e.g.* master regulators of cell fate; *Sucena et al., 2003*; *Colosimo et al., 2005*; *Chan et al., 2010*; *Yang et al., 2018*). One molecular feature predicted to be important for evolution is the ability of genes to produce multiple protein isoforms. A single protein-coding gene can produce multiple isoforms using alternative transcription initiation and termination sites combined with alternative splicing between exons (*Pan et al., 2008*; *Pal et al., 2011*). Isoform-specific evolution is found throughout vertebrates, including recent evolution of transcript expression in primates (*Barbosa-Morais et al., 2012*; *Merkin et al., 2012*; *Shabalina et al., 2014*; *Zhang et al., 2017*). Whether the increase in transcriptomic diversity is important for adaptive evolution remains an important question, and only a few examples have shown how isoform evolution could be involved in phenotypic diversity (*Mallarino et al., 2017*).

The ability of a gene to produce multiple protein isoforms might also play a role in the genesis of new genes. Over long evolutionary timescales, gene duplication and diversification can create paralogous genes with different functions (*Ohno, 1970*; *Innan and Kondrashov, 2010*). One central mystery in this process is the order of these two events; do mutations that duplicate genes occur first or does functional diversification preclude the duplication event? One mechanism the latter route can happen through is by gene sharing, or the ability of a gene to create multiple protein products (or a single protein product) that have two or more distinct functions (*Hughes, 1994*). If each isoform acts in different tissues or plays distinct roles in biological processes, subsequent duplication mutations can result in the separation of these isoforms into two distinct genes.

As a model for understanding the genetic basis of adaptive evolution in an animal model, we use the small nematode *Caenorhabditis elegans*. Besides its genetic tractability, use of this organism allows the analysis of evolution at different timescales. For example, experimental evolution can be used to study evolutionary processes in controlled environments on the order of 10–1000 generations (*Gray and Cutter, 2014*; *Teotónio et al., 2017*; *Penley et al., 2018*; *Chelo et al., 2019*; *Saxena et al., 2019*; *Wernick et al., 2019*). For longer timescales, a growing number of isolated and sequenced *Caenorhabditis* species can be used to study genetic differences responsible for species-level differences (*Ting et al., 2018*; *Yin et al., 2018*; *Bi et al., 2019*; *Stevens et al., 2019*).

For understanding short-term adaptation, we study two laboratory strains of *C. elegans*, called N2 and LSJ2, which descended from a single hermaphrodite isolated in 1951 (*Figure 1A*). These two lineages split from genetically identical populations between 1957 and 1958 and evolved in two very different laboratory environments – N2 grew on agar plates seeded with *E. coli* bacteria and LSJ2 in liquid cultures containing liver and soy peptone extracts (*McGrath et al., 2009*; *McGrath et al., 2011*; *Sterken et al., 2015*). By the time permanent means of cryopreservation were developed, approximately 300–2000 generations had passed, and ~ 300 new mutations arose and fixed in one of the two lineages (*McGrath et al., 2011*). Despite their genetic similarity, substantial divergence has occurred between these strains in terms of phenotype and fitness, including a large number of developmental, behavioral, and reproductive traits. Use of these strains allow us to identify causal genetic variants responsible for phenotypic and fitness changes. To date, five de novo, causal genetic variants have been identified in either the N2 or LSJ2 lineage (*de Bono and Bargmann, 1998*; *McGrath et al., 2009*; *Persson et al., 2009*; *McGrath et al., 2011*; *Duveau and Félix, 2012*; *Large et al., 2016*; *Large et al., 2017*; *Zhao et al., 2018*).

One of these mutations is an LSJ2-derived, 60 bp deletion at the 3' end the *nurf-1* gene that reduces growth rate, slows reproductive output, and prevents development into the dauer diapause state in response to ascaroside pheromones (*Figure 1B*) (*Large et al., 2016*). This genetic variant is beneficial in the LSJ2 liquid cultures in which it arose and fixed, but places animals at a disadvantage in the agar plate environments in which N2 evolved, an example of gene-environment interaction

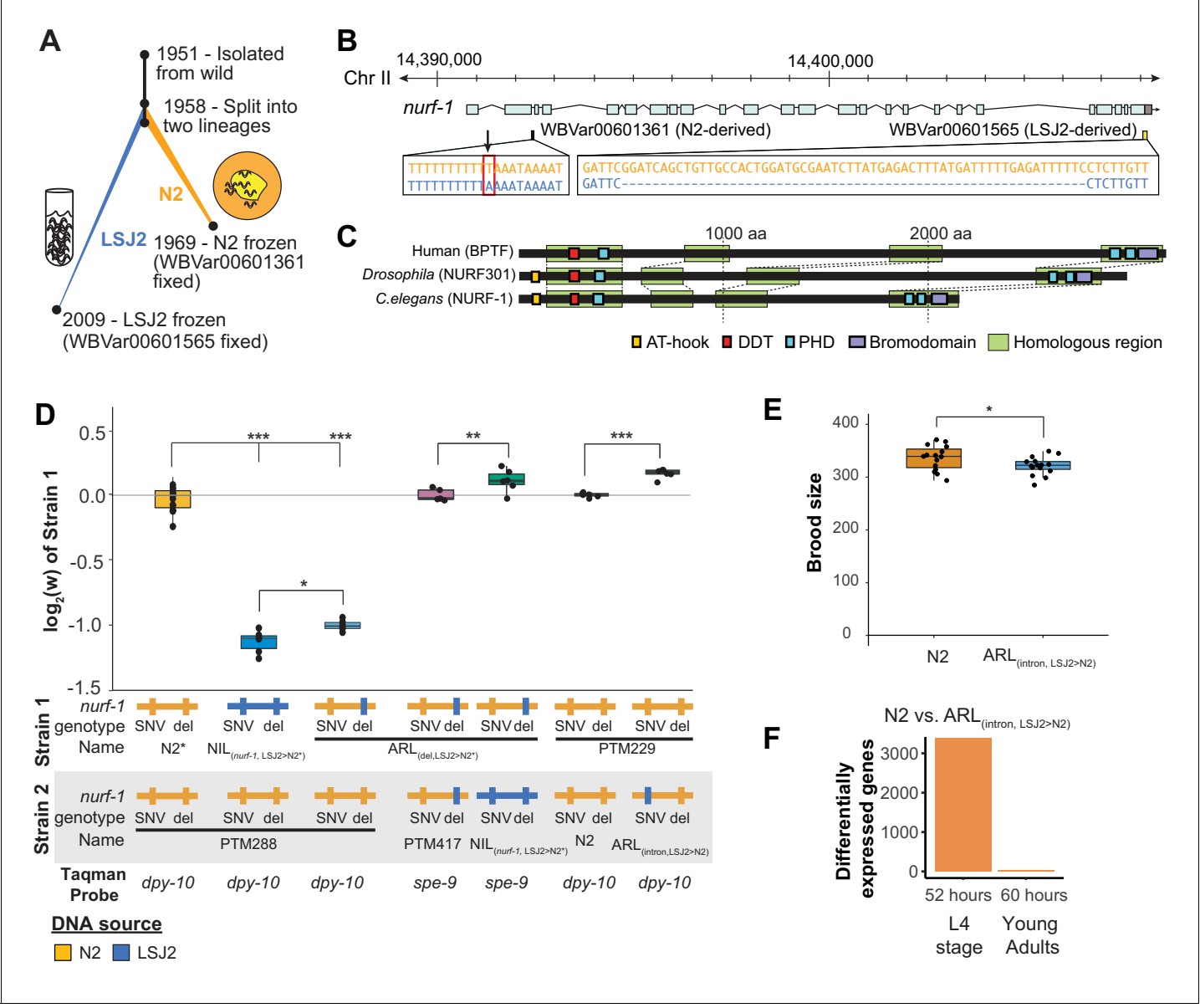

**Figure 1.** An N2-derived genetic variant in the intron of *nurf-1* increases fitness in laboratory conditions. (**A**) History of two laboratory adapted *C. elegans* strains N2 and LSJ2, which descend from the same individual hermaphrodite isolated in 1951. The N2 and LSJ2 lineage split sometime around 1958. N2 grew on agar plates with *E. coli* OP50 as a food source for around 11 years until they were cryopreserved. LSJ2 animals were cultured in liquid axenic media containing sheep liver extract and soy extract peptone as a food source for about around 51 years until they were cryopreserved. 302 genetic variations were fixed between these two strains, including two that fall in the *nurf-1* gene – WBVar00601361 and WBVar00601565. (**B**) Genetic location of two *nurf-1* variations. WBVar00601361 (in red box) is an N2-derived intron single nucleotide substitution T/A (N2/ancestral) in the 2nd intron of *nurf-1*. WBVar00601565 is an LSJ2-derived 60 bp deletion in the 3' end of *nurf-1* that removes the last 18 amino acids and part of the 3'-UTR. (**C**) Comparison of NURF-1 orthologs from *Drosophila* and humans showing position of protein domains and conserved regions as determined by Blastp and Clustal Omega. (**D**) Boxplot of pairwise evolutionary fitness differences between the indicated strains measured by directly competing the indicated strains against each other for five generations. PTM288 and PTM229 were created from the N2* and N2 strains, respectively, by a engineering DNA barcode in the *dpy-10* gene. PTM417 is the same genotype as ARL$_{(del\_LSJ2>N2)}$, with the exception of a background mutation in the *spe-9* gene that occurred during the construction of the ARL$_{(del\_LSJ2>N2)}$ strain (for details see Methods). This mutation was crossed out of the PTM417 strain and used as a barcode for the digital PCR reaction. The genotype of each *nurf-1* allele (shown in **B**) is indicated by color. The NIL strain also contains LSJ2 alleles of additional linked mutations, which is indicated by the blue horizontal line. (**E**) Total brood size of the N2 and ARL$_{(intron,LSJ2>N2)}$ strains. (**F**) Number of differentially expressed genes between synchronized N2 and ARL$_{(intron,LSJ2>N2)}$ animals harvested 52 hr (L4 stage - when spermatogenesis is active) or 60 hr (young adults - when oogenesis is active) after hatching. For all figures, each dot represents an independent replicate, the box indicates the interquartile values of all data, and the line indicates the median of all data. Positive values indicate strain one is more fit than strain two. Negative

*Figure 1 continued on next page*

*Figure 1 continued*

values indicate strain two is more fit than strain one. For all figures, n.s. indicates p>0.05, one star indicates significant difference at p<0.05 level, two stars indicate significant difference at p<0.01 level, and three stars indicate significant difference at p<0.001 level.

DOI: https://doi.org/10.7554/eLife.48119.002

The following source data and figure supplements are available for figure 1:

**Source data 1.** Source data for *Figure 1*.

DOI: https://doi.org/10.7554/eLife.48119.006

**Figure supplement 1.** Egg-laying rate of four strains.

DOI: https://doi.org/10.7554/eLife.48119.003

**Figure supplement 1—source data 1.** Source data for *Figure 1—figure supplement 1*.

DOI: https://doi.org/10.7554/eLife.48119.004

**Figure supplement 2.** Transcriptional analysis of N2 and $ARL_{(intron, LSJ2>N2)}$ at 52 and 60 hr.

DOI: https://doi.org/10.7554/eLife.48119.005

(*Large et al., 2016*). We proposed that *nurf-1* is a regulator of life-history tradeoffs. Life history tradeoffs represent competing biological traits requiring large energetic investments, such as the tradeoff between energy required for reproduction versus the energy required for individual survival. The difference in fitness of this allele in the two laboratory environments is potentially determined by how the life-history tradeoffs map into reproductive success.

Studies of *nurf-1* and its orthologs provide fundamental support for its role as a life history regulator. *nurf-1* encodes an ortholog of mammalian BPTF, a subunit of the NURF chromatin remodeling complex (*Barak et al., 2003*) (*Figure 1C*). *BPTF* encodes a large protein containing a number of domains that facilitate recruitment of NURF to specific regions of the genome for chromatin remodeling (*Alkhatib and Landry, 2011*), including domains that interact with sequence-specific transcription factors and three PHDs and a bromodomain that facilitate interactions with modified nucleosomes (*Li et al., 2006*; *Wysocka et al., 2006*; *Kwon et al., 2009*; *Ruthenburg et al., 2011*). Through its DDT domain (*Fyodorov and Kadonaga, 2002*), BPTF cooperates with ISWI to slide nucleosomes along DNA, changing access of promoter regions to transcription factors that drive gene transcription. In mammals, BPTF regulates cellular differentiation and homeostasis of specific cell-types and tissues, including the distal visceral endoderm (*Landry et al., 2008*), ecoplacental cone (*Goller et al., 2008*), hematopoietic stem/progenitor cells (*Xu et al., 2018*), mammary stem cells (*Frey et al., 2017*), T-cells (*Wu et al., 2016*), and melanocytes (*Koludrovic et al., 2015*). In *Drosophila*, the ortholog to BPTF, NURF301, regulates the heat shock response, pupation, spermatogenesis, and innate immunity (*Badenhorst et al., 2002*; *Badenhorst et al., 2005*; *Kwon et al., 2008*; *Kwon et al., 2009*). Many of these traits can be viewed as life-history tradeoffs, *e.g.* large energetic investments in individual survival through the development of the immune system vs. energetic transfers to offspring in the placenta or mammary glands. The evolution of BPTF/NURF-1 function might also be relevant in human disease. Genetic alterations in *BPTF* have been reported in tumors, including gene amplification and point mutations (*Buganim et al., 2010*; *Balbás-Martínez et al., 2013*). In addition, BPTF has been shown to be required for the transcriptional activity of c-MYC, a major human oncogene (*Richart et al., 2016*).

In this paper, we continue our studies of the evolution of the N2/LSJ2 laboratory strains. We demonstrate that an independent, beneficial mutation in the *nurf-1* gene was fixed in the N2 lineage, suggesting that *nurf-1* is a preferred genetic target for laboratory adaptation. To understand why *nurf-1* might be targeted, we explored the in vivo role in *C. elegans* development by taking advantage of CRISPR-Cas9 to test causal relationships that inform laboratory evolution and fitness effects. Our work suggests that the large, full-length isoform of *nurf-1*, primarily studied in mammals, is dispensable for development. Instead, two, largely non-overlapping isoforms are both essential for reproduction, having opposing effects on cellular differentiation of gametes into sperm or oocytes. Our results suggest that the ability of *nurf-1* to regulate life history tradeoffs is the result of exquisite regulation of NURF function through the balance of two competing isoforms, reminiscent of the principle of Yin and Yang. Finally, we demonstrate that these two isoforms have split into separate genes in a clade of related nematodes, potentially resolving transcriptional and functional conflict between the Yin and Yang isoforms transcription and function. Our work demonstrates how

evolution of isoforms can precede the origin of a new gene, supporting a role for gene sharing in the origin of functionally novel proteins.

## An N2-derived variant in the second intron of *nurf-1* increases fitness and brood size in laboratory conditions

We previously mapped differences in a number of traits (including reproductive rate, fecundity, toxin and anthelmintic sensitivity, and laboratory fitness) between N2 and LSJ2 to a QTL centered over *nurf-1,* which contains a derived mutation in both the N2 and LSJ2 lineages (*Figure 1A and B*) (*Large et al., 2016*; *Large et al., 2017*; *Zhao et al., 2019*). The LSJ2 allele of *nurf-1* contains a 60 bp deletion in the 3' end of the coding region of the gene, overlapping the stop codon and probably resulting in the translation of parts of the 3' UTR. The N2 allele of *nurf-1* contains an SNV that converts an A to a T in a homopolymer run of Ts in the $2^{nd}$ intron (*Figure 1B*). Using CRISPR-Cas9-based genome editing, we previously demonstrated that the LSJ2-derived deletion accounted for a large portion of the trait variance in reproductive rate explained by the QTL. However, it did not explain the entire effect of this locus (*Large et al., 2016*). We decided to test whether this additional genetic variant or variants affected fitness of the animals in laboratory conditions using a previously described pairwise competition assay (*Zhao et al., 2018*). To do so, we took advantage of three strains we had previously created; CX12311 is a near isogenic line with ancestral (non-N2) alleles of *npr-1* and *glb-5* crossed into an otherwise N2 genetic background, which we have used to eliminate the fitness and phenotypic effect of derived (N2) alleles of *npr-1* and *glb-5* (*Zhao et al., 2018*) (*Figure 1D* - referred to as N2*), NIL$_{(nurf-1,LSJ2>N2*)}$ is a near isogenic line containing LSJ2 alleles of both *nurf-1* mutations backcrossed into an N2* background (*Large et al., 2016*), and ARL$_{(-del,LSJ2>N2*)}$ is an allelic replacement line containing the LSJ2-derived 60 bp deletion edited into the N2* strain using CRISPR-Cas9. Phenotypic differences between the NIL$_{(nurf-1,LSJ2>N2*)}$ and ARL$_{(-del,LSJ2>N2*)}$ strains are caused by the N2-derived intron SNV in *nurf-1,* or one of the additional seven linked LSJ2-N2 genetic variants near *nurf-1.*

We measured the relative fitness of the N2*, NIL$_{(nurf-1,LSJ2>N2*)}$, and ARL$_{(del,LSJ2>N2*)}$ strains against PTM288, a version of N2* that also contains a silent mutation in the *dpy-10* gene (*Figure 1D*). The *dpy-10* silent mutation provides a common genetic variant that can be used to quantify the relative proportion of each strain on a plate using digital droplet PCR. Both the NIL$_{(nurf-1,LSJ2>N2*)}$ and ARL$_{(-del,LSJ2>N2*)}$ strains showed dramatically reduced fitness comparing to PTM288, consistent with our previous report showing that the 60 bp deletion is deleterious on agar plates (*Large et al., 2016*). However, the NIL$_{(nurf-1,LSJ2>N2*)}$ was quantitatively and significantly less fit than the ARL$_{(del,LSJ2>N2*)}$ strain, suggesting additional genetic variant(s) in the NIL$_{(nurf-1,LSJ2>N2*)}$ strain further reduced its fitness. To confirm this result, we also directly competed the NIL$_{(nurf-1,LSJ2>N2*)}$ and ARL$_{(del,LSJ2>N2*)}$ strains against each other, using a nearly neutral background mutation in *spe-9(kah132)* to distinguish the two strains (*Figure 1D*).

To determine if the N2-derived intron SNV in *nurf-1* (*Figure 1B*) was responsible for the fitness gains (as opposed to one of the seven linked LSJ2/N2 variants), we used CRISPR-Cas9 to directly edit the LSJ2 allele of the intron SNV into the standard N2 strain to create a strain we will refer to as ARL$_{(intron,LSJ2>N2)}$. We measured the relative fitness of the ARL$_{(intron,LSJ2>N2)}$ and N2 strains against PTM229 (a strain which again contains a *dpy-10* silent mutation). The ARL$_{(intron,LSJ2>N2*)}$ strain was significantly less fit than the N2 strain at a level similar to the difference between the NIL$_{(nurf-1,LSJ2>N2*)}$ and ARL$_{(del,LSJ2>N2)}$ strains (*Figure 1D*). These results indicate that beneficial alleles of *nurf-1* arose in both laboratory lineages - the 60 bp deletion makes LSJ2 animals more fit in liquid, axenic media (*Large et al., 2016*), and the intron SNV makes N2 animals more fit on agar plates seeded with bacteria.

In *C. elegans,* brood-size of hermaphrodites is negatively correlated to the timing of initial egg-laying. It has been suggested that this life-history tradeoff has been optimized in N2 animals (*Hodgkin and Barnes, 1991*; *Cutter, 2004*). After sexual maturation, gonads in the hermaphroditic sex initially undergo spermatogenesis before transitioning to oogenesis; a concomitant lengthening of spermatogenesis time increases the total brood size of hermaphrodites but also delays when reproduction can start. When we compared the total fecundity produced by the N2 and ARL$_{(intron,LSJ2>N2)}$ strains, we found a significant difference, with the ARL$_{(intron,LSJ2>N2)}$ strain producing ~ 30 fewer offspring than N2 (*Figure 1E*). This could indicate that spermatogenesis occurs for a longer period of time in N2 animals (to produce more sperm), or, alternatively, indicate that sperm are less

likely to fertilize an egg in the ARL$_{(intron,LSJ2>N2)}$ strain. The reproductive rate of the N2 and ARL$_{(intron, LSJ2>N2)}$ strains was largely unchanged throughout their reproductive lifespan (*Figure 1—figure supplement 1*).

RNA-seq analysis identified transcriptional differences caused by the intron SNV during spermatogenesis, supporting our hypothesis that sperm development is affected by this SNV. We collected RNA from synchronized N2 and ARL$_{(intron,LSJ2>N2)}$ hermaphrodites at two timepoints, 52 and 60 hr after hatching, which occur during spermatogenesis (52 hr) or oogenesis (60 hr). Interestingly, a large number of genes are differentially expressed between the two strains but only during the 52 hr timepoint (3384 genes vs. 25 genes) (*Figure 1F*, *Figure 1—figure supplement 2A*, and *Supplementary file 1*). Inspection of these differentially-expressed genes in a single-cell RNAseq dataset (*Cao et al., 2017*) demonstrated that although a portion of these 3384 genes are expressed in the germline, these genes are also expressed in additional tissues (*Figure 1—figure supplement 2B*). Gene ontology analysis suggests that cuticle development and innate immune responses are regulated by *nurf-1* (*Supplementary file 2*) consistent with the role of its orthologs in regulating immunity and melanocyte proliferation in *Drosophila* and humans (*Kwon et al., 2008*; *Landry et al., 2011*; *Koludrovic et al., 2015*; *Wu et al., 2016*). The restriction of most of these transcriptional changes to a specific timepoint (i.e. 52 hr) could reflect a specific-role for *nurf-1* in regulating genes undergoing short bursts of transcriptional upregulation during this developmental timepoint (e.g. molting), a specific-role for *nurf-1* in regulating cell number or activity of specific cell-types that are transiently present during this timepoint (e.g. spermatocytes), or some combination of both. These results suggest that the intron SNV regulates a number of developmental processes including spermatogenesis, molting, and innate immunity.

## *nurf-1* produces multiple transcripts encoding multiple protein isoforms

Our results suggest that selection acted repeatedly on *C. elegans nurf-1* during laboratory growth. The molecular nature of NURF-1, an essential subunit of the NURF chromatin remodeling complex, is surprising for a hotspot gene. In general, chromatin remodelers are thought of as ubiquitously expressed regulators with little variation in different cell types, akin to general function RNA polymerase proteins or ribosomes. Why would genetic perturbation of *nurf-1* lead to increased fitness? One potential clue is the complexity of the *nurf-1* locus. Previous cDNA analysis of *nurf-1* identified four unique transcripts encoding four unique isoforms (*Andersen et al., 2006*), two of which have been shown to affect different phenotypes (summarized in *Table 1*).

To identify other transcripts produced by *nurf-1* and quantify the relative proportions of each that are produced, we analyzed previously published Illumina short-read (*Brunquell et al., 2016*) (isolated from synchronized L2 larval animals) and Oxford Nanopore long-read RNA sequencing reads (*Roach et al., 2019*) (isolated from mixed populations) (*Figure 2—figure supplements 1–2*). Our results support many of the conclusions of *Andersen et al. (2006)* but contain a few surprises. We

**Table 1.** Summary of major *nurf-1* transcripts identified in *C. elegans*.

| Name | Evidence | | Size | | Conserved[c] | Predicted biological role in C. elegans[d] | Other names |
|------|-----------|------------|-----|-----|-----------|----------------------------|-------------|
| | Transcript[a] | Protein[b] | aa | kD | | | |
| *nurf-1.a* | N | - | 2197 | 252 | M,D | None | Full-length |
| *nurf-1.b* | C,N,I | W | 1621 | 186 | D | Reproduction, vulval development | N-terminal or NURF-1.A |
| *nurf-1.d* | C,N,I | W | 816 | 92 | - | Size, dauer, reproduction, axon guidance | C-terminal or NURF-1.C |
| *nurf-1.f* | C,N,I | W | 581 | 58 | - | None | NURF-1.E |
| *nurf-1.q* | N,I | - | 243 | 36 | - | None | - |

[a] C indicates full-length cDNA have been isolated for this transcript, N indicates evidence from direct sequencing of RNA or cDNA using Oxford Nanopore reads support this transcript, and I indicates evidence from Illumina short read RNA-seq supports this transcript

[b] W indicates evidence for the protein isoform was obtained using western blot

[c] M or D indicates an analogous isoform is described in mammals (mice or humans) or *Drosophila*, respectively

[d] Predictions from *Andersen et al. (2006)*, *Large et al. (2016)*, or *Mariani et al. (2016)*

DOI: https://doi.org/10.7554/eLife.48119.007

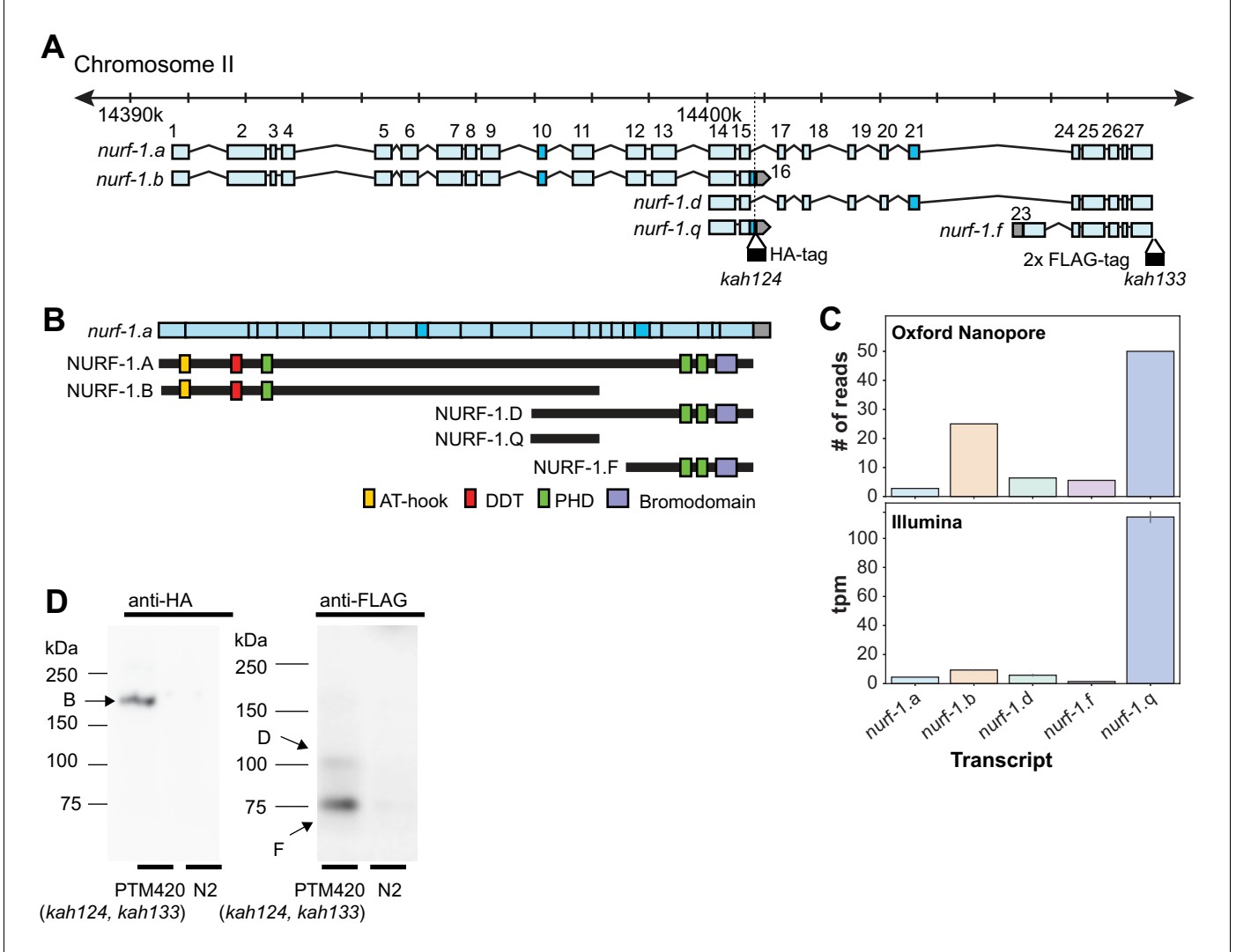

**Figure 2.** *nurf-1* encodes five major transcripts. (A) Genomic position of the five *nurf-1* transcripts supported by Illumina short read and Oxford Nanopore long reads. Each blue box is an exon. Exon number is indicated on the figure. Dark blue exons (10, 16, and 21) are alternatively spliced, resulting in a 6–9 bp difference in length (see *Figure 2—figure supplement 1* for details). Genomic location of the HA and FLAG epitope tag insertion site are shown in black along with their associated allele names. (B) The predicted protein isoforms produced by each of the five major transcripts and along with the domains each isoform contains. Immunoblots only supported translation of the B, D, and F isoforms (see panel D for details). For reference, the spliced *nurf-1.a* transcript is also shown. (C) Relative expression levels of each transcript, determined by number of Oxford Nanopore reads from a mixed population (top panel) or analysis of Illumina short reads from L2 staged animals using kallisto (bottom reads). tpm = transcripts per million. (D) Western blots of N2 and PTM420 strains. PTM420 contains the HA and FLAG epitope tags shown in panel A. Anti-HA antibody detected a band matching the expected size of the NURF-1.B isoform (arrow). Anti-FLAG antibody detected bands matching the expected size of the NURF-1.D and NURF-1.F isoforms (arrows).

DOI: https://doi.org/10.7554/eLife.48119.008

The following source data and figure supplements are available for figure 2:

**Figure supplement 1.** RNA-seq analysis of *nurf-1*.
DOI: https://doi.org/10.7554/eLife.48119.009

**Figure supplement 1—source data 1.** Source data for *Figure 2—figure supplement 1*.
DOI: https://doi.org/10.7554/eLife.48119.010

**Figure supplement 2.** *nurf-1* encodes multiple transcripts.
DOI: https://doi.org/10.7554/eLife.48119.011

**Figure supplement 2—source data 1.** Source data for *Figure 2—figure supplement 2*.
DOI: https://doi.org/10.7554/eLife.48119.012

**Figure supplement 3.** Relative expression levels of each *nurf-1* transcript, determined by analysis of Illumina short reads using kallisto (bottom reads).
*Figure 2 continued on next page*

*Figure 2 continued*

DOI: https://doi.org/10.7554/eLife.48119.013

**Figure supplement 4.** Identification of alternative BPTF species in human cancer cells.

DOI: https://doi.org/10.7554/eLife.48119.014

identified five major transcripts (*Figure 2A*) - three previously isolated (*nurf-1.b*, *nurf-1.d,* and *nurf-1. f*) but also two newly identified (*nurf-1.a* and *nurf-1.q*) (mapping of transcript names used in Andersen et al. are listed in *Table 1*). *nurf-1.a* encodes a full-length 2197 amino acid isoform analogous to the primary isoform of BPTF in humans and NURF301 in *Drosophila* (*Figure 1C*). Despite the expectation that *C. elegans* would produce a similar protein, the Oxford Nanopore long-read data are the only evidence supporting its existence. The *nurf-1.q* transcript is predicted to produce a 243 amino acid unstructured protein. With the exception of the full-length *nurf-1.a* transcript, the overlap of these transcripts is quite minimal, resulting in predicted isoforms with unique protein domains and functions (*Figure 2B*).

We quantified the relative expression of these five transcripts by either counting the number of Nanopore reads that matched the transcript or by using kallisto (*Bray et al., 2016*) to predict transcript abundance using Illumina short-read sequencing data (*Figure 2C*). These predictions qualitatively agreed in transcript ranking of expression strength (although quantitative variation in predictions were observed, reflective of the different technologies or developmental stages of the animals). Surprisingly, the newly described *nurf-1.q* transcript was the most highly expressed followed by the *nurf-1.b* transcript, and the *nurf-1.a*, *nurf-1.d* and *nurf-1.f* were expressed at similar lower levels.

Although each of the five major transcripts are transcribed, this result does not necessarily mean they are translated into stable protein products. To facilitate analysis of NURF-1 proteins, we used CRISPR-Cas9 to fuse two distinct epitope tags (HA and 3xFLAG tag) to the endogenous *nurf-1* locus, just prior to the stop codons in the 16[th] and 28[th] exon, respectively (*Figure 2A*). Immunoblot analysis supported the expression of the B, D, and F isoforms, but not the A or Q isoforms (*Figure 2D*). Although larger proteins, such as the A isoform, can be difficult to transfer during immunoblots, the lack of a band matching the small Q isoform suggests the highly expressed *nurf-1.q* transcript is not translated into protein or the protein is rapidly degraded.

Based upon these results, we speculated that the intronic SNV, which we have shown regulates total fecundity and fitness in laboratory conditions (*Figure 1*), could specifically alter the expression level of the *nurf-1.b* transcript. However, analysis of all five *nurf-1* transcript levels, using the previously described RNA-seq data on the N2 and ARL$_{(intron,LSJ2>N2)}$ strains, did not reveal any significant expression differences (*Figure 2—figure supplement 3*). Potentially the effect of this SNV has cell-type specific effects in spermatocytes, however, our data, using RNA collected from the whole animal, does not allow us to test this hypothesis.

We also investigated whether similar isoforms could be expressed in human cells, using western blots on a small panel of human cancer cell lines. Interestingly, besides a band matching the expected size of the canonical full-length isoform, a number of additional bands were observed between 150–250 kD (*Figure 2—figure supplement 4A*). Using mass-spectrometry, we confirmed the presence of multiple BPTF peptides in the bands detected by western blotting (*Figure 2—figure supplement 4*), consistent with one or more of these bands representing novel BPTF isoforms. Potentially, these isoforms could play a role in cancer metastasis, although we provide no such evidence here. Despite the presence of these additional bands, the full-length version of BPTF is the most highly expressed isoform (*Figure 2—figure supplement 4D*), consistent with its importance in mammalian species.

## The B and D isoforms are both essential for reproduction and the F isoform modifies the heat shock response

Genetic analysis of *nurf-1* primarily relied on two deletion alleles, *n4293* and *n4295* (*Figure 3A*) (*Andersen et al., 2006*). The *n4293* allele deletes the first exon and predicted transcriptional start site of the *nurf-1.a* and *nurf-1.b* transcripts. The *n4295* allele deletes three exons of the *nurf-1.a*, *nurf-1.d*, and *nurf.1.f* transcripts that encode a C-terminal PHD domain (*Figure 3—figure*

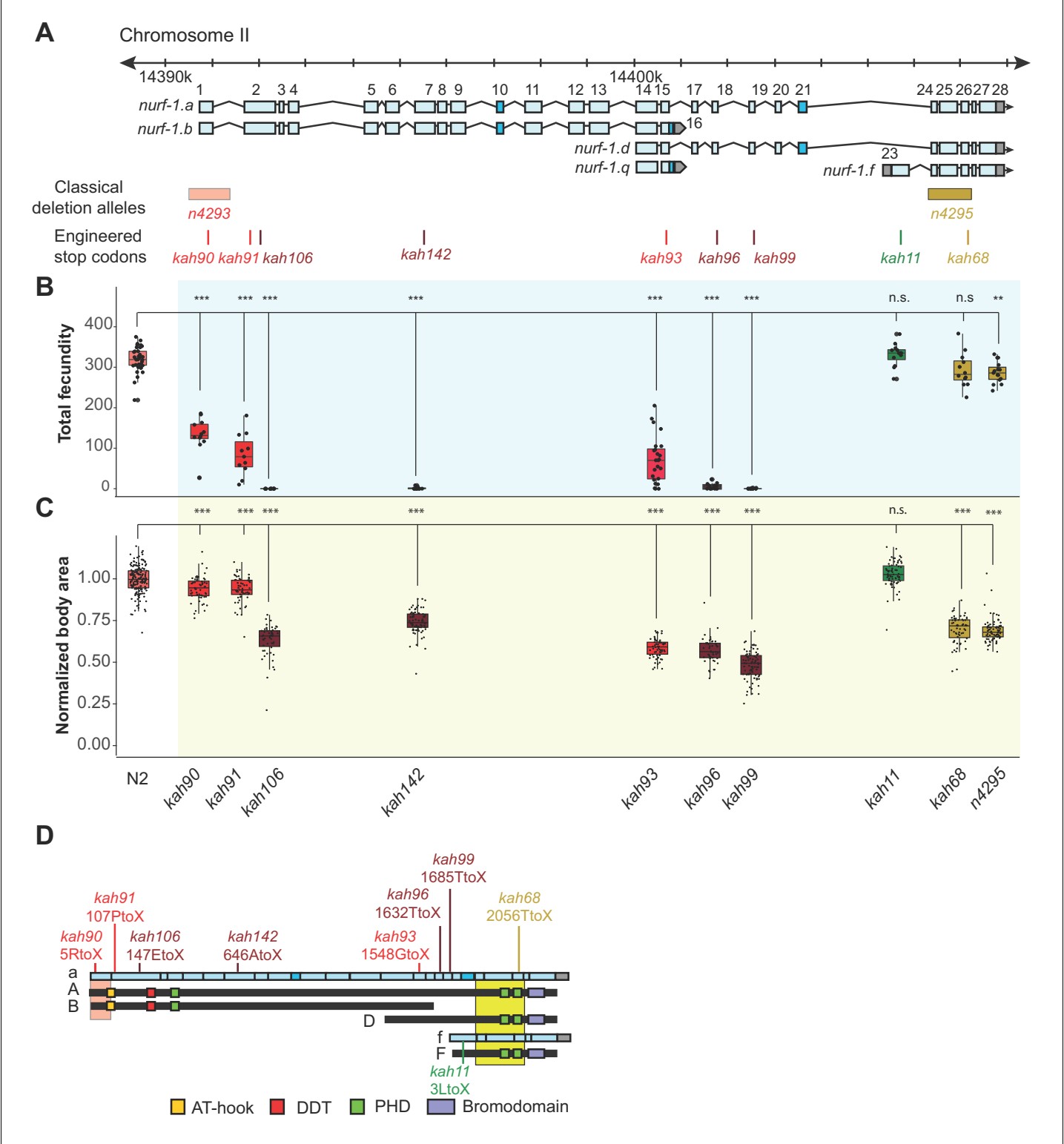

**Figure 3.** An additional isoform besides NURF-1. B is necessary for reproduction in *C. elegans*. (**A**) Genomic positions of *nurf-1* classical deletion alleles and nine engineered stop codons created using CRISPR/Cas9 based gene editing. Each allele is color-coded by the reproductive ability of homozygous strains. Green is statistically indistinguishable from wild-type, yellow indicates slightly reduced brood size and change in reproductive rate, red indicates substantially reduced brood size in the first generation and eventual sterility after multiple generations of homozygosity, and dark red indicates sterility in the first generation of homozygosity. (**B**) Fecundity of indicated strains (shown in x-axis of panel) (**C**) Normalized body area of the indicated strains. Normalized body area was calculated by thresholding video recordings of each strain to segment individual animals and registered
*Figure 3 continued on next page*

*Figure 3 continued*

throughout each frame of the video. Each dot represents the average area of a single worm, normalized to the N2 data. For red or dark red strains (panel A), measurements were taken on animals homozygous for a single generation. (D) Predicted amino acid change of engineered stop codons and classical alleles on the NURF-1 isoforms. The *kah11* mutation only affects the F isoform.

DOI: https://doi.org/10.7554/eLife.48119.015

The following source data and figure supplements are available for figure 3:

**Source data 1.** Source data for *Figure 3*.

DOI: https://doi.org/10.7554/eLife.48119.019

**Figure supplement 1.** Histone recognition domains in NURF-1.D are not essential for its activity.

DOI: https://doi.org/10.7554/eLife.48119.018

**Figure supplement 2.** Reproductive output of indicated strains at indicated times.

DOI: https://doi.org/10.7554/eLife.48119.016

**Figure supplement 3.** Fecundity analysis of the indicated alleles of *nurf-1*.

DOI: https://doi.org/10.7554/eLife.48119.017

*supplement 1*) necessary for human BPTF function. Comparison of the phenotypes of the *n4293* and *n4295* homozygotes leads to the model that the B isoform is essential for reproduction and the A, D, and/or F isoforms have subtle effects on growth rate and reproductive rate (*Table 1*).

To further delineate the biological role of each isoform, we used CRISPR-Cas9 to engineer nine stop codons in eight exons of the *nurf-1* gene: the first, second (two positions), 7th, 15th, 18th, 19th, 23rd, or 26th exons (*Figure 3A*). The predicted effects of these stop codons on each major isoform are shown in *Figure 2—figure supplement 4* and *Table 2*. Homozygous animals for each mutation were assayed for total brood size and growth rate. Analysis of the phenotypes of these mutants indicated that our working model was incorrect. Instead, we propose that both the B and D isoforms are essential for reproduction.

As expected, engineering stop codons in the first, second, and 7th exons greatly reduced fecundity, resulting in either sterility, or a mortal germline phenotype, initially reducing total brood size of animals, before eventually causing complete sterility after around three-to-five generations of homozygosity (*Figure 3B and C*). Although the qualitative phenotypes of these four alleles agreed, we observed interesting quantitative differences between them. The second stop codon in the second exon (*kah106*) and the stop codon in the 7th exon (*kah142*) reduced growth and fecundity more than the first exon stop codon (*kah90*) or the first stop codon in second exon (*kah91*) (*Figure 3B and C*). We suspect this result indicates the presence of an internal ribosome entry site in the middle of the second exon at the 122nd Methionine, causing the expression of two isoforms from a single transcript. The reduced severity of the first two stop codon alleles can be explained by their inability to affect the protein sequence of the second isoform. An alternative possibility is a difference in frequency of translational read-through of each stop codon, which are interpreted as sense codons at a low frequency (*Jungreis et al., 2011*).

Unexpectedly, engineering stop codons in the 18th and 19th exons also caused a mortal germline phenotype (*kah96* and *kah99*) (*Figure 3B*). This result was surprising, because the *n4295* allele, predicted to be a loss-of-function allele for the D and F isoforms due to the loss of the PHD and bromo-domains, does not have a mortal germline phenotype. We excluded a number of potential explanations for this discrepancy. A suppressor for the *n4295* allele could have fixed during the construction of this strain. However, the *kah68* allele, which contains a stop codon within the *n4295*

**Table 2.** Predicted effect of stop codon mutations on NURF-1 isoforms.

| Isoform | kah90 | kah91 | kah106 | kah142 | kah93 | kah96 | kah99 | kah11 | kah68 | Length |
|---------|-------|-------|--------|--------|-------|-------|-------|-------|-------|--------|
| NURF-1.A | 5R | 107P | 147E | 646A | 1548G | 1632T | 1685Q, 1689P, 1693N | | 2056T | 2197 |
| NURF-1.B | 5R | 107P | 147E | 646A | 1548G | - | - | | - | 1621 |
| NURF-1.D | - | - | - | | 170G | 254T | 307Q, 311P, 315N | | 675T | 816 |
| NURF-1.F | - | - | - | | - | - | - | 3L | 440T | 581 |
| NURF-1.Q | - | - | - | | 170G | - | - | | - | 243 |

DOI: https://doi.org/10.7554/eLife.48119.020

deleted region, phenocopies the *n4295* allele and not the *kah96* and *kah99* animals (*Figure 3B and C*, and *Figure 3—figure supplement 2*)). Another possibility is that the D isoform suppresses the F isoform; loss of both isoforms (in the *n4295* background) is tolerated, but loss of just the D isoform (in the *kah96* or *kah99* backgrounds) allows the F isoform to prevent reproduction. However, we could exclude this possibility as the double mutant containing both the *n4295* allele and the 18th exon stop allele phenocopied the *kah96* single mutant (*Figure 3—figure supplement 3*). Additionally, specific loss of the F isoform by the 23rd exon stop allele (*kah11*) did not affect the phenotype of animals (*Figure 3B and C*). Our data suggest that, unlike human BPTF, the ability of NURF-1 to bind modified histones is not required for its function. We further confirmed this hypothesis by editing conserved residues in these the PHD and bromodomains necessary for recognition of the H3K4me3 and H4K16ac marks (*Figure 3—figure supplement 1*).

The most parsimonious explanation of our data is that either the A or D isoform is essential for reproduction in *C. elegans*. Compound heterozygote tests allowed us to distinguish between these possibilities, indicating that the D isoform is required for reproduction and wild-type growth rate, and the A isoform is dispensable for reproduction and development (*Figure 4*). We first verified that the *kah93*, *kah96*, and *kah106* alleles were recessive by measuring the fecundity of heterozygous animals (*Figure 4B*). Next, we examined the fecundity of *kah106/kah96* compound heterozygotes, which are predicted to lack only the A isoform, due to the production a single unaffected copy of the B isoform from the *kah96* haplotype and a single unaffected copy of the D isoform from the *kah106* haplotype. If the A isoform was essential for reproduction, we would expect these compound heterozygotes to be sterile or have severe defects in fecundity. However, these animals were indistinguishable from wild-type, suggesting that the full-length A isoform is not essential (*Figure 4B*). The *kah106/kah93* compound heterozygotes showed similar results. These animals are predicted to encode one unaffected copy of the D isoform, one truncated copy of the B isoform, and zero unaffected copies of the A isoform. These animals were mostly wild-type, with a small reduction in total fecundity (*Figure 4B*). We interpret this to mean that the A isoform is not essential and the truncation of the B isoform slightly perturbs its function, causing a slight reduction in fecundity. Finally, we analyzed *kah93/kah96* compound heterozygotes. These animals are predicted to encode zero wild-type copies of the D isoform, one wild-type copy of the B isoform, and zero wild-type copies of the A isoform. These animals were essentially sterile. Taken together, we conclude that the B and the D isoform are both essential for reproduction.

To confirm that the D isoform is essential, we also created a transgenic strain containing an integrated construct driving a *nurf-1.d* cDNA from its endogenous promoter. This transgene could fully rescue the fecundity phenotype of the *kah96* allele and partially rescue the fecundity phenotype of the *kah93* allele (*Figure 4C*). This transgene could also rescue the reproductive timing and fecundity changes of the *n4295* allele and the LSJ2-derived 60 bp deletion (*kah3*) (*Figure 4C* and *Figure 4—figure supplement 1*). As expected, this transgene could not rescue the *kah106* allele, which creates a stop codon in the B isoform. These data further support a requirement of both the B and D isoforms for reproduction.

Although the F isoform does not seem to have an effect in normally developing animals, it is involved in the heat shock response. Multiple reports have demonstrated that *nurf-1* is upregulated in response to heat shock (*Brunquell et al., 2016*; *Li et al., 2016*). By analyzing RNA-seq reads from these two papers, we found that the *nurf-1.f* transcript was specifically upregulated in both datasets, with increased coverage of the 23rd exon as well as the 24th through 28th exons (*Figure 4—figure supplement 2A and B*). We confirmed that the increased transcription of the *nurf-1.f* transcript also increased NURF-1.F protein abundance (*Figure 4—figure supplement 2C and D*). Transcriptional analysis of strains lacking the F isoform indicated that the initial transcriptional response to heat shock was largely the same, but the long-term transcriptional response of a subset of genes was affected (*Figure 4—figure supplement 2E–G*). We conclude that the F isoform is specifically up-regulated by heat shock and plays a modulatory role in determining the long-term transcriptional response to heat shock.

## The B and D isoforms have opposite effects on cell fate during gametogenesis

Although the B and D isoforms are both required for reproduction, the molecular mechanism that these isoforms operate through could be different. One possibility is that the long-form of NURF-1

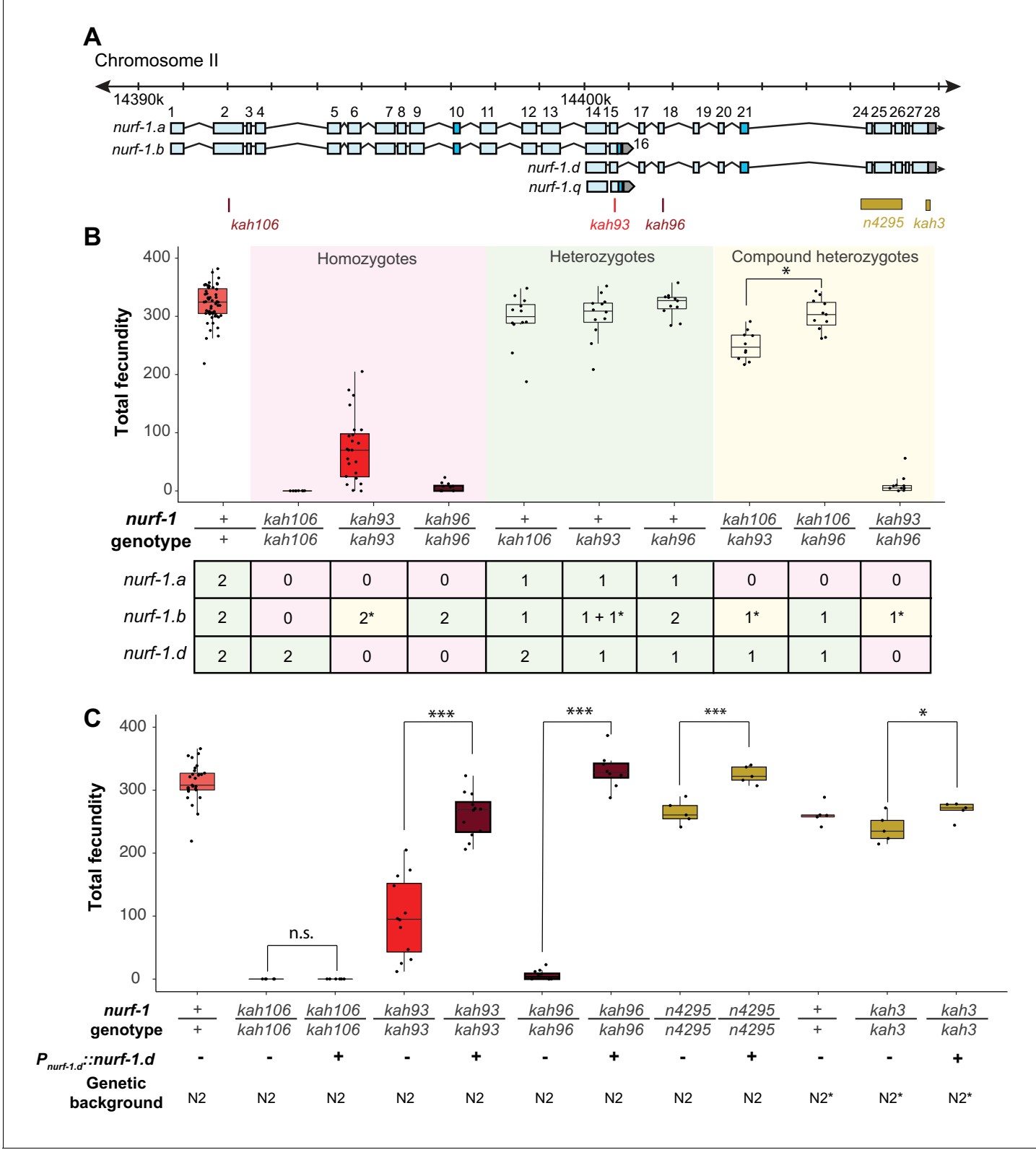

**Figure 4.** Genetic analysis suggests the NURF-1. B and NURF-1.D isoforms are essential for reproduction in *C. elegans*. (**A**) Genomic positions of stop codon or classical deletion mutations used for compound heterozygote or transgenic rescue analysis of B and C. *kah3* is a CRISPR/Cas9 genomic edit of the LSJ2-derived 60 bp deletion. (**B**) Fecundity of homozygote (red), heterozygote (green), and compound heterozygote mutants (yellow) as indicated in the x-axis. The table below the x-axis is the predicted effect of each mutant strain on the indicated *nurf-1* isoforms. The number in the table

*Figure 4 continued on next page*

*Figure 4 continued*

indicates the number of functional copies. The star indicates the milder predicted effect of *kah93* on NURF-1.B, as it only truncates 73 of 1621 amino acids. The y-axis shows the fecundity for each strain. (C) Fecundity of indicated strains with and without the presence of an integrated *nurf-1.d* transgene. The genetic background is also indicated. N2* contains ancestral introgressions of the *npr-1* and *glb-5* genes.

DOI: https://doi.org/10.7554/eLife.48119.021

The following source data and figure supplements are available for figure 4:

**Source data 1.** Source data for *Figure 4*.

DOI: https://doi.org/10.7554/eLife.48119.024

**Figure supplement 1.** Egg-laying rate of *n4295* and ARL(del, LSJ2>N2*) transgenic *nurf-1.d* cDNA rescue.

DOI: https://doi.org/10.7554/eLife.48119.022

**Figure supplement 2.** Heat shock specifically upregulates NURF-1.F.

DOI: https://doi.org/10.7554/eLife.48119.023

has split into two subunits - both isoforms participate as part of the NURF complex, cooperating together to regulate reproduction. However, the D isoform might instead modify NURF activity by competing for binding with transcription factors or regions of the genome to which NURF is recruited. A third possibility is that the D isoform acts through a NURF-independent pathway.

To gain insights into the molecular nature of the D isoform, we decided to determine precisely how the B and D isoforms regulate reproduction, using three *nurf-1* stop alleles (*Figure 5A*). For hermaphrodites to produce a fertilized egg, the gonads must produce both male and female gametes at different developmental times (*Figure 5B*). Initially, gametogenesis produces sperm, creating approximately 300 sperm at which point a permanent sperm-to-oocyte switch occurs. From this time, gametogenesis produces oocytes until the animal dies or the gonad ceases to function (*Hubbard and Greenstein, 2005*). A number of defects could cause sterility – inability to form gametes, inability to create sperm, inability to create oocytes, or defects in the sperm and/or oocyte function. We used DAPI staining to characterize the production of sperm and oocytes in three *nurf-1* mutants (*Figure 5C and D*). We first tested *kah106* mutants, which lack the B isoform (*Figure 5A*), for the ability to produce sperm. Compared with N2 animals, which create ~ 300 sperm per animal, the number of sperm produced by *kah106* animals was greatly reduced, resulting in the production of only approximately 60 sperm (*Figure 5D*). These animals produced a normal number of oocytes, indicating that spermatogenesis seemed to be affected specifically (*Figure 5E*). We interpret these data as evidence that hermaphrodites that lack the NURF-1.B isoform spend less time in spermatogenesis before transitioning to oogenesis. We next tested *kah96* mutants which lack the D isoform. These animals produced approximately 500 sperm and almost no oocytes (*Figure 5C-E*). We interpret these data as evidence that hermaphrodites that lack the D isoform are unable to transition from spermatogenesis to oogenesis. Finally, we performed similar experiments on *kah93* mutants, which lack the D isoform and have a truncated B isoform. These animals showed an intermediate phenotype, with normal number of sperm but reduced number of oocytes (*Figure 5D and E*). The reduced activity of the B isoform due to its truncation potentially allows other factors to transition the animals to oogenesis, resulting in the milder defects found in the *kah93* animals (*Figure 3B*).

Although animals that lack either the B or D isoform are unable to reproduce, the cause of sterility is different at the cellular level. To further study the molecular effects of perturbing *nurf-1* function, we transcriptionally profiled adult N2*, NIL(nurf-1,LSJ2>N2*), ARL(del, LSJ2, N2*), and LSJ2 animals, which contain various combinations of the N2 and LSJ2-derived *nurf-1* mutations (*Supplementary file 1*). A multi-dimensional scaling plot indicated that the N2* and ARL(del) replicates formed two unique clusters, and the LSJ2 and NIL(nurf-1) replicates largely overlapped in a third cluster (*Figure 5—figure supplement 1A*). The genetic variation surrounding the *nurf-1* locus is responsible for the majority of transcriptional differences between adult LSJ2 and N2* animals, suggesting most of the fixed variants do not have a dramatic effect on transcription on N2-like growth conditions. Although the LSJ2-derived 60 bp deletion regulates transcription, additional genetic variation in the NIL(nurf-1) strain, presumably from the N2-derived intron variant, also regulates transcription in adult animals.

To study the effects of the 60 bp deletion and intron SNV on transcription, we focused on two comparisons: 1) the N2* vs ARL(del, LSJ2>N2*), which will identify transcriptional changes caused by the 60 bp deletion and 2) the NIL(nurf-1, LSJ2>N2*) vs ARL(del, LSJ2>N2*), which will identify transcriptional

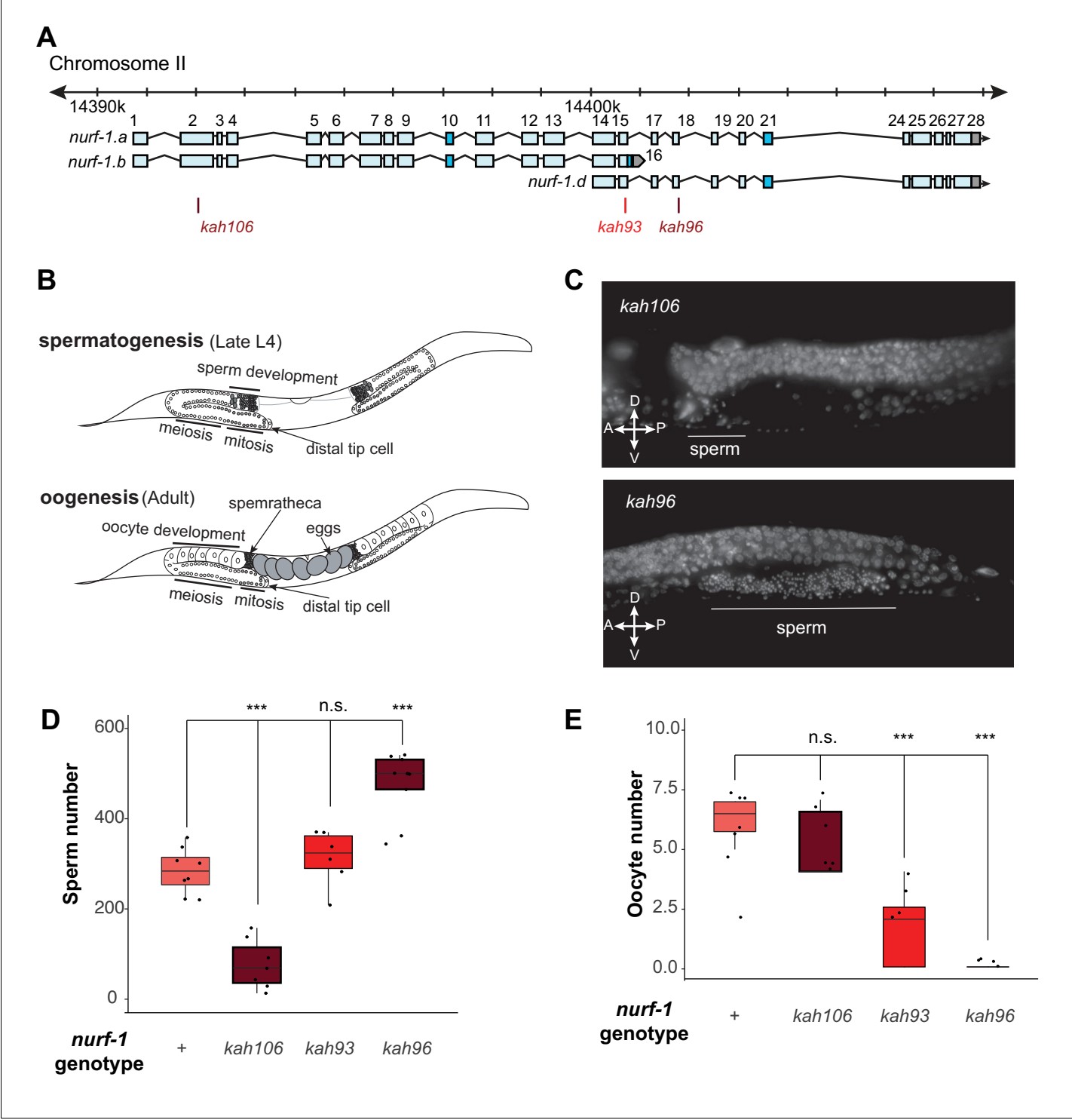

**Figure 5.** NURF-1.B and NURF-1.D have opposite effects on the sperm-to-oocyte switch in hermaphrodites. (**A**) Genomic position of the previously-described stop codon mutants used in B and C. (**B**) Summary of gametogenesis of *C. elegans*. Animals undergo spermatogenesis during the late L4 and then transition to oogenesis stage during maturation to adulthood. The number of sperm produced during spermatogenesis can be determined by counting sperm in the spermatheca when oogenesis has begun. (**C**) Representative fluorescence images of one spermatheca for DAPI stained young adult animals. Each tiny dot represents the condensed chromosomes of a single sperm. (**D**) Sperm number of indicated strains. L4 animals were synchronized and allowed to develop for an additional 12 hr. DAPI staining was used to identify and count the number of sperm in each animal. Each dot represents a single animal. (**E**) Oocyte number of indicated strains. L4 animals were synchronized and allowed to develop for an additional 12 hr. DAPI staining was used to identify and count the number of oocytes in each animal.

*Figure 5 continued on next page*

*Figure 5 continued*

DOI: https://doi.org/10.7554/eLife.48119.025

The following source data and figure supplement are available for figure 5:

**Source data 1.** Source data for *Figure 5*.

DOI: https://doi.org/10.7554/eLife.48119.027

**Figure supplement 1.** Transcriptome analysis of strains containing N2/LSJ2 genetic variation linked to *nurf-1*.

DOI: https://doi.org/10.7554/eLife.48119.026

changes caused by the intron SNV (as well as linked mutations in the NIL other than the 60 bp deletion). We expect that the latter comparison will mostly report the changes of the intron SNV, as it accounts for most of the fitness differences between the two strains. We observed a positive correlation between these two comparisons (*Figure 5—figure supplement 1B*). The most parsimonious explanation for this observation is that both the N2 and LSJ2-derived alleles in *nurf-1* regulate the activity of a common molecular target, which is likely to be the NURF complex.

## A duplication in a sister clade of *Caenorhabditis* species creates two separate *nurf-1* genes

Previous work in *C. briggsae* characterized the role of *nurf-1* in reproduction, including the isolation of *nurf-1* cDNAs in this species (*Chen et al., 2014*). Interestingly, although transcripts matching the *nurf-1.b* and *nurf-1.d* were isolated from this species, they no longer shared any exons with each other, suggesting that they were expressed from two separate genes (*Figure 6A*). Further, spliced leader sequences to the 5' end of both transcripts matched *sl1* sequence, suggesting that these two genes were not expressed as a single operon (*Blumenthal, 2012*). We compared the gene products using BLAST and found that the shared exons in *C. elegans* had duplicated in *C. briggsae*, with one set of each retained in each of the new genes (*Figure 6A*). Short-read transcriptomics data for this species matched the cDNA analysis; we found evidence for transcripts matching *nurf-1.b*, *nurf-1.d*, and *nurf-1.f* (*Figure 6—figure supplements 1–3*). Unlike *C. elegans*, *C. briggsae* seemed to have lost both the *nurf-1.a* and *nurf-1.q* transcripts.

Analysis of the *nurf-1* gene structure within the context of the *Caenorhabditis* phylogeny suggested that the exon duplication and separation of *nurf-1* into separate genes occurred at the base of a clade containing 11 described species, including *C. brenneri* and *C. tribulationis* (*Figure 6B*). We determined the *nurf-1* gene structure in 22 of the 32 *Caenorhabditis* species with published genomes and transcriptomes (*Kiontke et al., 2011*; *Stevens et al., 2019*) (*Figure 6—figure supplements 1–3*). Like *C. briggsae*, the species in the *brenneri/tribulationis* clade express a transcript matching *nurf-1.b* from a single gene (which we call *nurf-1–1*). These species also express two transcripts matching *nurf-1.d* and *nurf-1.f* from a second gene, called *nurf-1–2*. Analysis of the spliced leader sequence of the 5' end of the nurf-1.d transcript only identified *sl1* sequence, consistent with separation of these genes into distinct transcriptional units. None of these species appears to express *nurf-1.a* or *nurf-1.q* transcripts (*Figure 6—figure supplements 1–3*). RNA-seq data for species outside of this clade (*Figure 6—figure supplements 1–3*) matched the transcription pattern of *C. elegans*, suggesting that these species express five major transcripts from a single *nurf-1* gene: *nurf-1.a*, *nurf-1.b*, *nurf-1.d*, *nurf-1.f*, and *nurf-1.q*. These data suggest that the *C. elegans* transcript structure is ancestral.

Phylogenetic analysis of the duplicated ~ 200 amino acid sequence was used to evaluate different hypotheses surrounding the timing and number of duplication events. The analysis supported the model that the split of *nurf-1* into two distinct genes happened once within the common ancestor of the *brenneri/tribulationis* clade (*Figure 6C* – additional possible trees shown in *Figure 6—figure supplement 4*). The topology recovered for the region of *nurf-1* outside the duplication is congruent with the species tree (*Figure 6—figure supplement 5*). Interestingly, the rate of amino acid substitution in the duplicated region was accelerated in *nurf-1–1* relative to *nurf-1–2* (p<0.001; Welch's t-test) suggesting that this region experienced positive selection and/or relaxed selection after this duplication event occurred. Comparison of the synonymous vs. non-synonymous substitution rate in three closely-related species pairs was also consistent with an increase in the rate of protein evolution in the duplicated region following the separation of *nurf-1* into independent genes (*Table 3*).

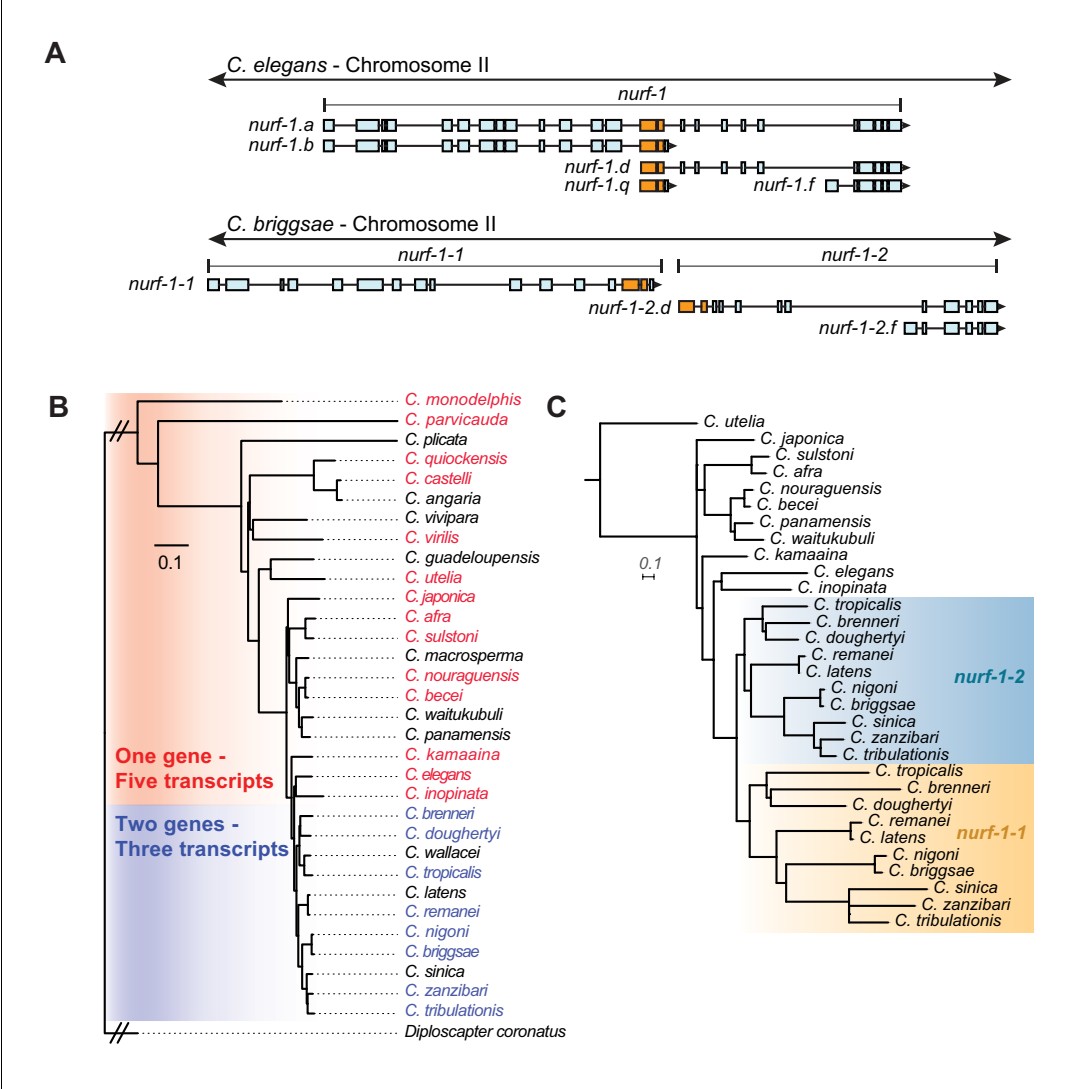

**Figure 6.** A duplication of the shared exons of the *nurf-1.b* and *nurf-1.d* transcripts resulted in the split of *nurf-1* it into two separate genes in a subclade of *Caenorhabditis* species. (**A**) Comparison of two species with different versions of *nurf-1*. In *C. elegans*, *nurf-1.b* and *nurf-1.d* overlaps in the 14th and 15th exon (shown in orange). In *C.briggsae*, a duplication of the orange exons resulted in separation of *nurf-1.b* and *nurf-1.d* into separate genes. *C. briggsae* has also lost expression of the *nurf-1.a* and *nurf-1.q* transcripts. (**B**) Distribution of the two versions of *nurf-1* (shown in panel **A**) in 32 Caenorhabditis species. Red indicates the *C. elegans* version, blue indicates the *C. briggsae* version, and black indicates a *nurf-1* version that could not be determined. The species phylogeny suggests that a duplication event occurred in the common ancestor of the *brenneri/tribulationis* clade. (**C**) The most well supported topology of the duplicated region is consistent with a single duplication event. Orange indicates protein sequence from the duplicated region in the *nurf-1–1* gene, and turquoise indicates protein sequence from the duplicated region in the *nurf-1–2* gene. Non-colored branches indicate unduplicated *nurf-1* sequence. The rate of amino acid substitution in the *nurf-1–1* duplicated region has also increased, as seen in the branch lengths. Scale is in substitutions per site.

DOI: https://doi.org/10.7554/eLife.48119.028

The following source data and figure supplements are available for figure 6:

**Source data 1.** Source data for *Figure 6*.
DOI: https://doi.org/10.7554/eLife.48119.035

**Figure supplement 1.** *nurf-1* isoform structure for 22 *Caenorhabditis* species.
DOI: https://doi.org/10.7554/eLife.48119.029

**Figure supplement 2.** Sashimi plots for *Caenorhabditis* species with one *nurf-1* gene.
DOI: https://doi.org/10.7554/eLife.48119.030

**Figure supplement 3.** Sashimi plots for *Caenorhabditis* species with two *nurf-1* genes.
DOI: https://doi.org/10.7554/eLife.48119.031

**Figure supplement 4.** Five hypothetical topologies related to the timing and number of duplication events involved in the *nurf-1* gene split.

*Figure 6 continued on next page*

*Figure 6 continued*

DOI: https://doi.org/10.7554/eLife.48119.032

**Figure supplement 5.** Maximum likelihood tree of the B isoform and *nurf-1–1*.

DOI: https://doi.org/10.7554/eLife.48119.033

**Figure supplement 6.** Maximum likelihood tree of the duplicated region of *nurf-1* in 22 species.

DOI: https://doi.org/10.7554/eLife.48119.034

## Discussion

In this paper, we make use of CRISPR/Cas9-enabled gene editing to characterize the *nurf-1* gene in *C. elegans* and then study the sequence and expression of *nurf-1* orthologs in other *Caenorhabditis* species. The combination of genetics and evolutionary analysis allowed us to make a number of surprising observations. First, we show that an SNV in the 2nd intron of *nurf-1* that fixed in the N2 laboratory strain increases the fitness and fecundity of the N2 strain. Second, we show that the full-length isoform of *nurf-1* has split into two essential, mostly non-overlapping isoforms with opposite effects on cell fate in differentiating gametes. Finally, we show that the B and D isoforms have split into separate genes in a subset of *Caenorhabditis* species. These data show that *nurf-1* can be genetically perturbed to increase fitness of animals in new environments and has experienced long-term evolutionary changes that have split its function and regulation into two isoforms/genes (*Figure 7A and B*).

### Evolution of NURF-1/BPTF across phyla

In humans and *Mus musculus*, an abundance of evidence confirms that the long-form isoform of *BPTF*, which is orthologous to *nurf-1*, is the primary isoform in the NURF chromatin remodeling complex (*Alkhatib and Landry, 2011*). While a subset of *BPTF* exons are alternatively spliced, these events will not lead to the large changes in size we observe in the *nurf-1* gene. One exception is the FAC1 isoform, which encompasses 801 N-terminal amino acids of BPTF (*Bowser et al., 1995*). While FAC1 is found in amyloid Alzheimer's patients and enriched in the nervous system (*Bowser et al., 1995*; *Landry et al., 2008*), a biological role for this isoform has not been described. FAC1 is smaller and lacks conserved protein sequence found in the B isoform of *nurf-1*, suggesting an independent evolutionary origins and function.

In *Drosophila*, an intermediate state between humans and nematodes is found. Two major isoforms of NURF301 (the ortholog to *nurf-1*) have been described: a full-length form of NURF301 analogous to the full-length mammalian BPTF and an N-terminal form of NURF301 analogous to the NURF-1.B isoform of *C. elegans*. Both isoforms form NURF complexes and regulate gene expression (*Kwon et al., 2009*). Genetic analysis suggests that full-length NURF301 is required for

**Table 3.** dN/dS ratio in three *Caenorhabditis* species pairs.

| Sp. pair | Duplication[a] | *nurf-1–1* or *nurf-1.b* | | | | *nurf-1–2* or *nurf-1.d* | | |
| --- | --- | --- | --- | --- | --- | --- | --- | --- |
| | | Dup. Reg.[b] | Other[c] | Ratio[d] | | Dup. Reg.[b] | Other[c] | Ratio[d] |
| *C. afra/ C. sulstoni* | N | 0.136[e] | 0.121 | 1.1 | | 0.116[e] | 0.072 | 1.6 |
| *C. nigoni/ C. briggsae* | Y | 0.249 | 0.085 | 2.9 | | 0.111 | 0.019 | 5.8 |
| *C. remanei/ C. latens* | Y | 0.295 | 0.121 | 2.4 | | 0.177 | 0.048 | 3.7 |

[a] Duplication indicates whether the species pair contain the duplicated exons that create two *nurf-1* genes

[b] Dup. Reg. indicates dN/dS was calculated using the region of the alignment that contains the duplication

[c] Other indicates dN/dS was calculated using the region of the alignment that does not contain the duplication

[d] Ratio was calculated by dividing the dN/dS value of the Dup. Reg. by the Other

[e] The dN/dS values for the *nurf-1.b* and *nurf-1.d* in the duplicated region were different due to the b transcript encoding two additional amino acids in the 14th exon (before the M initiation codon in the d isoform) and the amino acids encoded by the 16th alternatively spliced exon.

DOI: https://doi.org/10.7554/eLife.48119.036

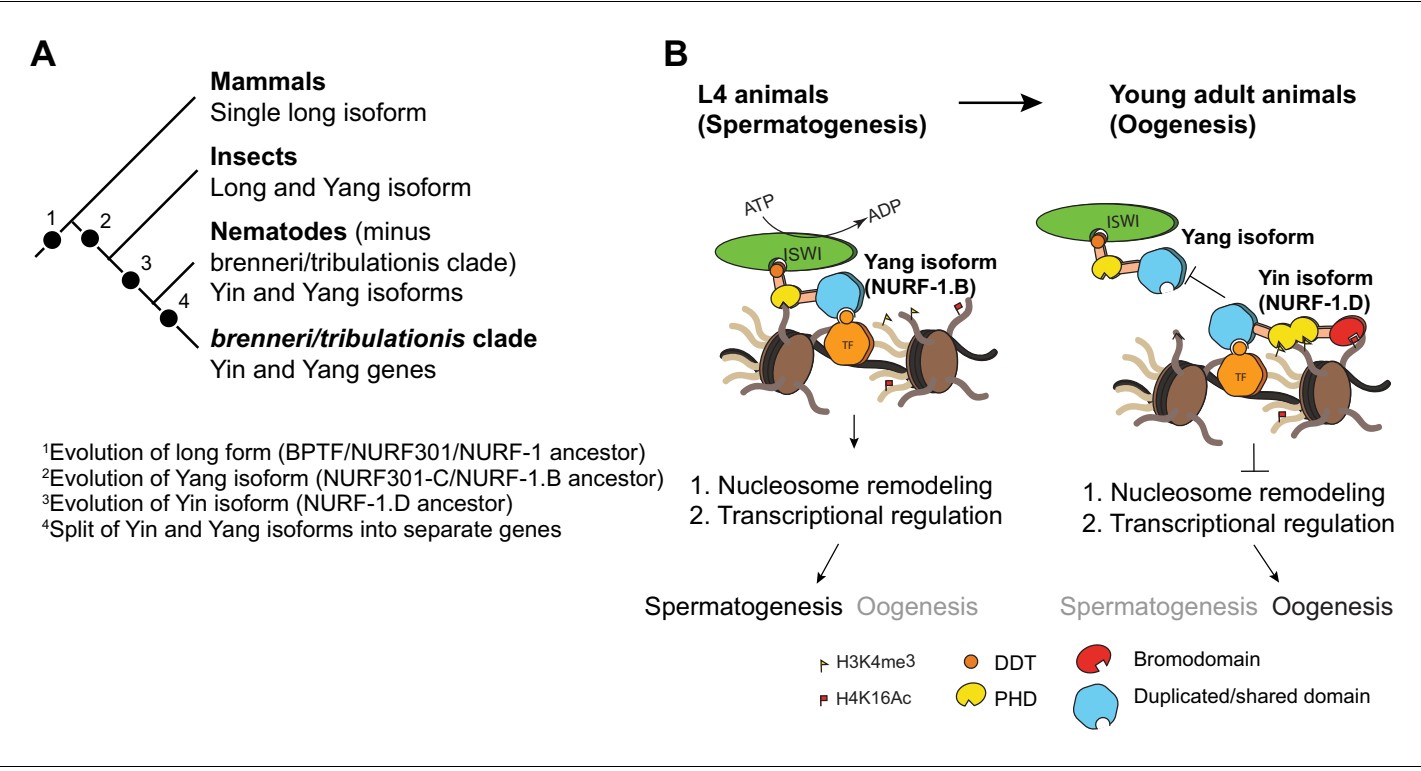

**Figure 7.** Proposed antagonistic (Yin-Yang) working model of two *nurf-1* isoforms in *C. elegans*. (A) Descriptive phylogeny with proposed major transitions in *nurf-1* isoform evolution. Each dot indicates the timepoint of a major *nurf-1* isoform evolution event. (B) Proposed molecular mechanism for NURF-1 isoforms. The NURF-1.B isoform interacts with ISWI through its DDT domain to form a NURF complex capable of remodeling chromatin at specific regions of the chromosome. NURF is recruited to these regions through interactions with specific transcription factors using protein domains encoded by the overlapping exons. This remodeling is necessary for transcriptional responses for spermatogenesis. Due to some unknown signal, after spermatogenesis has resulted in the production of ~ 300 sperm, the NURF-1 D isoform outcompetes the NURF complex away from its target loci, causing the loss of transcription of key spermatogenesis genes, resulting in gametogenesis transitioning from spermatogenesis to oogenesis. The binding affinity of PHD domains and bromodomainto histone strengthens this repression, but they are not completely necessary for the ability of the D isoform to outcompete the B isoform.

DOI: https://doi.org/10.7554/eLife.48119.037

gametogenesis in both sexes while the N-terminal isoform is required for regulation of pupation and innate immunity.

Nematodes have retained the N-terminal isoform but seem to have lost use of the full-length isoform for most biological traits (*Andersen et al., 2006*). Instead, they express two C-terminal isoforms (D and F) that appear to be a recent evolutionary innovation, likely occurring before the origin of the *Caenorhabditis* lineage. We show that the D isoform (or the Yin isoform) is essential in *C. elegans*, and seems to act in opposition to the B isoform (or the Yang isoform) to regulate the sexual fate of differentiating gametes. The requirement of two antagonistic isoforms (the B and D) for reproduction is reminiscent of the principle of Yin and Yang. Genetic pathways often include both positive and negative regulators of transcription and ultimate phenotype, however, rarely are both the factors encoded by the same genetic locus. While there is growing appreciation of isoform-specific regulation of many genes, *nurf-1* appears to be unusually complex in this regard (although not unprecedented – see *Müller and Basler, 2000*; *Berry et al., 2001*; *Wang et al., 2009*).

We propose a molecular mechanism to explain the actions of the B and D isoforms to regulate transcription (*Figure 7B*). These two isoforms share 207 amino acids of protein sequence, which falls in a region that is thought to facilitate physical interactions with transcription factors (*Alkhatib and Landry, 2011*). NURF-1.B participates as part of the NURF complex, which is recruited to certain promoters by binding to transcription factors. At these loci, NURF promotes or represses expression of target genes by remodeling the chromatin surrounding promoters and gene bodies. For unknown

molecular reasons, NURF-1.D preferentially binds to these transcription factors, displacing the NURF complex from these genomic regions, causing a change of chromatin state and gene expression.

## Microevolution of NURF-1/BPTF

We showed that independent, beneficial alleles in *nurf-1* were fixed in two laboratory strains of *C. elegans* that each experienced an extreme shift in environment from their natural habitats. The N2-derived SNV results in the change of a run of homopolymers in the 2nd intron of the *nurf-1.b* transcript. Such a change could act as an enhancer for the *nurf-1.d* promoter, but the nature of the genetic change and position is more consistent with a role in regulating the *nurf-1.b* transcript. Analysis of RNA sequencing data did not identify any obvious changes in levels of any of the *nurf-1* transcripts and it is unclear by what molecular mechanism it regulates *nurf-1* activity. Potentially, it could increase pausing of the RNA polymerase at the homopolymer run or could regulate RNA splicing by changing the secondary structure of the RNA molecule. In general, such a mutation would not be predicted by most bioinformatic approaches to have a phenotypic effect. Only the low genetic diversity between the LSJ2 and N2 strains allowed us to focus on this variant, and eventually demonstrate this particular variant is causal.

The probability of two beneficial mutations happening in both lineages by random chance is quite small. Less than 300 genes (out of ~ 20,000 total) harbor derived mutations in either the N2 or LSJ2 lineage (*McGrath et al., 2011*). Only a handful of these fixed mutations are expected to be beneficial; our recent QTL mapping of fitness differences on agar plates only identified the *nurf-1* locus (*Zhao et al., 2019*) and the small effective population sizes (~4–100) are expected to lead to the fixation of a number of nearly-neutral mutations through genetic drift and draft. Our work suggests *nurf-1* is a genetic target for adaptation to the extreme changes in environments associated with laboratory growth.

Targeting of *nurf-1* is consistent with its role as a regulator of life history tradeoffs. Many traits influence individual and offspring survival; however, the mapping of these traits onto fitness is thought to be dependent on the environmental niche an organism occupies. The LSJ2-derived deletion in *nurf-1* modified life history tradeoffs to prioritize individual survival over reproduction; by shunting energy away from reproduction and growth, they increased their chances of surviving on poor, unnatural food. N2 animals grew on agar plates seeded with *E. coli* bacteria, which they can readily consume and metabolize into a useful energy source. In these conditions, survival is not the primary concern; each animal has three days to eat as much food as possible and produce as many progenies as possible to maximize the probability one of their offspring is transferred to the new food source. It is reasonable to think that the N2 and LSJ2 laboratory conditions represent opposite extremes along a life history axis encompassing individual survival and reproduction. The N2 mutation favors reproduction while the LSJ2 mutation favors survival.

In humans, genetic alterations in *BPTF* have been reported in several types of cancer and a role of BPTF in transcriptional regulation by c-MYC has been demonstrated, in agreement with its chromatin-binding function (*Richart et al., 2016*). Using well-characterized and validated antibodies against BPTF, we found several molecular species with unexpected electrophoretic mobilities in human cancer cells (*Figure 2—figure supplement 4*). Using mass-spectrometry, we confirmed the presence of multiple BPTF peptides in the bands detected by western blotting (*Figure 2—figure supplement 4*). These findings raise the possibility that these protein sequence variants have non-canonical functions. Given that stress adaptation is a hallmark of cancer - allowing tumor cells to survive and evolve following Darwinian selection processes - and the role of *nurf-1* in *C. elegans* demonstrated here, it is tempting to speculate a role for such diversity of isoforms in the life histories of cancer cells. However, our work simply shows that additional forms of BPTF exist. Whether they have a biological role still needs to be determined.

## Split of nurf-1 *into separate genes potentially resolves conflict between the Yin and Yang isoforms caused by shared exons*

In a clade of *Caenorhabditis* nematodes, the *nurf-1* gene has split into two separate genes, an example of gene birth resulting in the duplication of a portion of the *nurf-1* gene. Multigene families are common in most species and protein domains are often shuffled between genes. While the importance of gene duplication is not controversial, the exact mutational events and evolutionary forces

responsible for the fixation of independent genes with different functions is less understood. Here we seem to have uncovered an example of how gene sharing, specifically through the creation of unique isoforms, can contribute to this process. In the lineage that led to the *C. elegans species*, *nurf-1* first evolved changes in isoform use, resulting in the creation and essential action of the *nurf-1.d* transcript, and the loss of essentiality of the long *nurf-1.a* transcript. In this case, partitioning of the biological function and protein domains in each *nurf-1* isoform created diversification of protein products.

What are the evolutionary forces responsible for the split of *nurf-1* into two genes? One possibility is developmental system drift. Under this scenario, the separation of the two isoforms into two distinct genes does not signify any important evolutionary change in the function of the two genes. Neutral processes are responsible for the initial fixation of the duplication and the change does not provide any future evolutionary benefit.

However, there are a few additional possible ways adaptive evolution could play a role. First, correlated with the separation of the Yin and Yang transcripts into two genes is the loss of the full-length *nurf-1.a* and *nurf-1.q* transcripts. Loss of these transcripts could have provided a fitness benefit for animals. Consider, in order to produce both the *nurf-1.b* and *nurf-1.d* transcripts (i.e. the Yin and Yang transcripts) in the same cell, there must be a mechanism to distinguish between transcripts containing the 1st to 15th exons (the *nurf-1.b* transcripts) and transcripts initiating from the 14th exon (the *nurf-1.d* transcripts). In the former case, the 15th exon is spliced to the 16th exon to terminate the transcript. In the latter case, the 15th exon is spliced to the 17th exon, along with the remaining 3' exons. Alternatively, the cell might not distinguish between transcripts, but uses each alternative splice site at a constant ratio (i.e. 80% of the time, the 15th exon is spliced to the 16th exon and 20% of the time, the 15th exon is spliced to the 17th exon). In the latter scenario, two additional transcripts must be produced. Intriguingly, these two transcripts match *nurf-1.a* and *nurf-1.q*, suggesting these transcripts are non-functional biproducts of molecular conflict between *nurf-1.b* and *nurf-1.d*. Potentially, production of the *nurf-1.a* and *nurf-1.q* transcripts could come at an energetic cost.

Multiple lines of evidence are consistent with the *nurf-1.a* and *nurf-1.q* transcripts playing non-biological roles. First, while the *nurf-1.q* transcript is produced at high levels, we were unable to observe its product in our immunoblots, suggesting that it is either not translated or the protein product is rapidly degraded. Second, our genetic tests were unable to identify a biological role for *nurf-1.a*. Third, we observe a loss of both the *nurf-1.a* and *nurf-1.q* transcripts in the species that have split *nurf-1* into two genes. It would have been quite easy for these species to retain expression of *nurf-1.q* in their current configuration, either through a promoter in front of the 14th exon in the *nurf-1–1* gene, or an alternative stop exon after the 2nd exon of the *nurf-1–2* gene, since both of these elements existed in the ancestral state.

Second, duplication of the shared exons could facilitate future evolutionary change. Escape from adaptive conflict is a mechanism by which gene duplication can resolve the situation where a single gene is selected to perform multiple roles (*Des Marais and Rausher, 2008*). After duplication, each copy is free to improve its function independently. As organisms evolve, recruitment of NURF to specific loci could be accomplished by changing its binding with specific transcription factors through amino acid changes in NURF-1. The most rapidly evolving portion of the protein is within the 14th and 15th exons, suggesting positive selection acts on this region of the protein, potentially changing the transcription factors NURF-1 binds to. One issue that arises in species containing a single *nurf-1* gene is the pleiotropy of genetic changes in the shared region; changing the amino acid sequence of the B isoform also changes the D isoform. Are there situations where modifying one isoform but not the other is preferred? In the clade of nematodes that have duplicated *nurf-1*, each gene is free to evolve independently. We present evidence that in these species, the duplicated region is free to evolve more rapidly. It should be interesting to characterize the exact function of this duplicated region and determine if these changes in protein sequence facilitate changes in transcription factor binding in an adaptive manner.

## Conclusion

A fundamental problem in evolutionary biology is understanding the genetic mechanisms responsible for phenotypic diversity in extant species. Here, we present one route to address this problem. Experimental evolution and genetic analysis can be used to identify evolutionary relevant genes and understand their function. This knowledge can be leveraged to understand patterns of evolution of

these genes in other species. We believe that merging genetics, genomics, and molecular evolution is a powerful approach to understand the evolutionary mechanisms responsible for long-term adaptation and species level differences.

## Materials and methods

**Key resources table**

| Reagent type or resource | Designation | Source of reference | Identifiers | Additional information |
|---|---|---|---|---|
| Gene (*C. elegans*) | *nurf-1* | WormBase | Wormbase ID: WBGene00009180 | Sequence: F26H11.2 |
| Gene (human) | BPTF | National Center for Biotechnology Information | Gene ID: 2186 | |
| Strain, strain background (*E. coli*) | OP50 | Caenorhabditis Genetics Center (CGC) | RRID: WB-STRAIN:OP50 | |
| Strain (*C. elegans*) | CX12311 | PMID: 21849976 | RRID: WB-STRAIN:CX12311 | Strain Background: N2, Request a strain: please email the corresponding author |
| Strain (*C. elegans*) | PTM66 | PMID: 21849976 | | Strain Background: N2, Request a strain: please email the corresponding author |
| Strain (*C. elegans*) | PTM88 | PMID: 21849976 | | Strain Background: N2, Request a strain: please email the corresponding author |
| Strain (*C. elegans*) | PTM288 | PMID: 30328811 | RRID: WB-STRAIN:PTM288 | Strain Background: N2, Request a strain: please email the corresponding author |
| Strain (*C. elegans*) | PTM229 | PMID: 30328811 | RRID: WB-STRAIN:PTM229 | Strain Background: N2, Request a strain: please email the corresponding author |
| Strain (*C. elegans*) | PTM98 | This paper | RRID: WB-STRAIN:PTM98 | Strain Background: N2, Request a strain: please email the corresponding author |
| Strain (*C. elegans*) | PTM113 | This paper | RRID: WB-STRAIN:PTM113 | Strain Background: N2, Request a strain: please email the corresponding author |
| Strain (*C. elegans*) | PTM116 | This paper | RRID: WB-STRAIN:PTM116 | Strain Background: N2, Request a strain: please email the corresponding author |
| Strain (*C. elegans*) | PTM117 | This paper | RRID: WB-STRAIN:PTM117 | Strain Background: N2, Request a strain: please email the corresponding author |
| Strain (*C. elegans*) | PTM118 | This paper | RRID: WB-STRAIN:PTM118 | Strain Background: N2, Request a strain: please email the corresponding author |
| Strain (*C. elegans*) | PTM167 | This paper | RRID: WB-STRAIN:PTM167 | Strain Background: N2, Request a strain: please email the corresponding author |

*Continued on next page*

*Continued*

| Reagent type or resource | Designation | Source of reference | Identifiers | Additional information |
|---|---|---|---|---|
| Strain (*C. elegans*) | PTM170 | This paper | RRID: WB-STRAIN:PTM170 | Strain Background: N2, Request a strain: please email the corresponding author |
| Strain (*C. elegans*) | PTM189 | This paper | RRID: WB-STRAIN:PTM189 | Strain Background: N2, Request a strain: please email the corresponding author |
| Strain (*C. elegans*) | PTM203 | This paper | RRID: WB-STRAIN:PTM203 | Strain Background: N2, Request a strain: please email the corresponding author |
| Strain (*C. elegans*) | PTM211 | This paper | RRID: WB-STRAIN:PTM211 | Strain Background: N2, Request a strain: please email the corresponding author |
| Strain (*C. elegans*) | PTM316 | This paper | RRID: WB-STRAIN:PTM316 | Strain Background: N2, Request a strain: please email the corresponding author |
| Strain (*C. elegans*) | PTM317 | This paper | RRID: WB-STRAIN:PTM317 | Strain Background: N2, Request a strain: please email the corresponding author |
| Strain (*C. elegans*) | PTM319 | This paper | RRID: WB-STRAIN:PTM319 | Strain Background: N2, Request a strain: please email the corresponding author |
| Strain (*C. elegans*) | PTM322 | This paper | RRID: WB-STRAIN:PTM322 | Strain Background: N2, Request a strain: please email the corresponding author |
| Strain (*C. elegans*) | PTM325 | This paper | RRID: WB-STRAIN:PTM325 | Strain Background: N2, Request a strain: please email the corresponding author |
| Strain (*C. elegans*) | PTM332 | This paper | RRID: WB-STRAIN:PTM332 | Strain Background: N2, Request a strain: please email the corresponding author |
| Strain (*C. elegans*) | PTM354 | This paper | RRID: WB-STRAIN:PTM354 | Strain Background: N2, Request a strain: please email the corresponding author |
| Strain (*C. elegans*) | PTM371 | This paper | RRID: WB-STRAIN:PTM371 | Strain Background: N2, Request a strain: please email the corresponding author |
| Strain (*C. elegans*) | PTM372 | This paper | RRID: WB-STRAIN:PTM372 | Strain Background: N2, Request a strain: please email the corresponding author |
| Strain (*C. elegans*) | PTM373 | This paper | RRID: WB-STRAIN:PTM373 | Strain Background: N2, Request a strain: please email the corresponding author |
| Strain (*C. elegans*) | PTM376 | This paper | RRID: WB-STRAIN:PTM376 | Strain Background: N2, Request a strain: please email the corresponding author |

*Continued on next page*

*Continued*

| Reagent type or resource | Designation | Source of reference | Identifiers | Additional information |
|---|---|---|---|---|
| Strain (*C. elegans*) | PTM416 | This paper | RRID: WB-STRAIN:PTM416 | Strain Background: N2, Request a strain: please email the corresponding author |
| Strain (*C. elegans*) | PTM417 | This paper | RRID: WB-STRAIN:PTM417 | Strain Background: N2, Request a strain: please email the corresponding author |
| Strain (*C. elegans*) | PTM420 | This paper | RRID: WB-STRAIN:PTM420 | Strain Background: N2, Request a strain: please email the corresponding author |
| Strain (*C. elegans*) | PTM487 | This paper | RRID: WB-STRAIN:PTM487 | Strain Background: N2, Request a strain: please email the corresponding author |
| Strain (*C. elegans*) | PTM489 | This paper | RRID: WB-STRAIN:PTM489 | Strain Background: N2, Request a strain: please email the corresponding author |
| Strain (*C. elegans*) | PTM512 | This paper | RRID: WB-STRAIN:PTM512 | Strain Background: N2, Request a strain: please email the corresponding author |
| Strain (*C. elegans*) | PTM517 | This paper | RRID: WB-STRAIN:PTM517 | Strain Background: N2, Request a strain: please email the corresponding author |
| Cell line (Human) | Colo-205 | American Type Culture Collection (Rockville, MD) | | |
| Cell line (Human) | MCF-7 | American Type Culture Collection (Rockville, MD) | | |
| Cell line (Human) | MDA-MB-231 | American Type Culture Collection (Rockville, MD) | | |
| Cell line (Human) | Hela | American Type Culture Collection (Rockville, MD) | | |
| Cell line (Human) | A549 | G. Roncador, CNIO | | |
| Sequence-based reagents (Plasmid) | Plasmid: pSM | Cori Bargmann Lab (Rockefeller University) | | |
| Sequence-based reagents (Plasmid) | Plasmid: pDD162 PrU6::*dpy-10_sgRNA* | PMID: 27467070 | | CRISPR/Cas9 gene editing |
| Sequence-based reagents (Plasmid) | Plasmid: pDD162 Pr*eft3::Cas9* | PMID: 27467070 | | CRISPR/Cas9 gene editing |
| Sequence-based reagents (Plasmid) | Plasmid: pCFJ90 | Addgene | | http://www.wormbuilder.org/test-page/about-mossci/ |
| Sequence-based reagents (Plasmid) | Plasmid: pCFJ104 | Addgene | | http://www.wormbuilder.org/test-page/about-mossci/ |
| Sequence-based reagents (Plasmid) | Plasmid: pCFJ151 | Addgene | | http://www.wormbuilder.org/test-page/about-mossci/ |

*Continued on next page*

*Continued*

| Reagent type or resource | Designation | Source of reference | Identifiers | Additional information |
|---|---|---|---|---|
| Sequence-based reagents (Plasmid) | Plasmid: pCFJ601 | Addgene | | http://www.wormbuilder.org/test-page/about-mossci/ |
| Antibody | (mouse monoclonal) anti-HA | Life Technologies | Cat. No.: 326700 | (1:500) |
| Antibody | (mouse monoclonal) anti-DYKDDDDK | Life Technologies | Cat. No.: MA191878 | (1:1000) |
| Antibody | (mouse monoclonal) anti-FLAG | Millipore Sigma | Cat. No.: F3165 | (1:1000) |
| Antibody | Horseradish peroxidase-conjugated secondary antibodies | Dako Glostrup | | (1:10000) |
| Peptide, recombinant protein | BPTF | Novus Biologicals | Cat. No.: NB100-41418 | |
| Peptide, recombinant protein | Vinculin | Sigma | Cat. No.: V9131 | |
| Sequence-based reagents (Oligonucleotide) | *dpy-10 (cn64)* | PMID: 25161212 | | CRISPR/Cas9 gene editing |
| Commercial assay, kit | Taqman probe: *dpy-10 (kah82/kah83)* | Thermal: Custom TaqMan SNP Genotyping Assays | PTM09 | |
| Commercial assay, kit | NEB Q5 site directed mutagenesis kit | NEB | Cat. No.: E0554 | |
| Commercial assay, kit | Next Ultra II Directional RNA Library Prep Kit | NEB | Cat. No.: E7760S | |
| Commercial assay, kit | Zymo DNA isolation kit | Zymo | Cat. No.: D4071 | |
| Commercial assay, kit | Zymo DNA cleanup kit | Zymo | Cat. No.: D4064 | |
| Commercial assay, kit | ddPCR Supermix for Probes | BIORAD | Cat. No.: 1863010 | |
| Commercial assay, kit | Droplet Generation Oils | BIORAD | Cat. No.: 1863005 | |
| Commercial assay, kit | ddPCR Droplet Reader Oil | BIORAD | Cat. No.: 1863004 | |
| Commercial assay, kit | VECTASHIELD antifade Mounting Medium with DAPI | VECTOR | Cat. No.: H-1200 | |
| Software, Algorithm | edgeR | PMID: 19910308 | RRID:SCR_012802 | Opensource: https://bioconductor.org/packages/release/bioc/html/edgeR.html |
| Software, Algorithm | SARtools | PMID: 27280887 | RRID:SCR_016533 | Opensource: https://github.com/PF2-pasteur-fr/SARTools |
| Software, Algorithm | IGV | PMID: 21221095 | RRID:SCR_011793 | https://software.broadinstitute.org/software/igv/ |
| Software, Algorithm | Kallisto | PMID: 27043002 | RRID:SCR_016582 | https://pachterlab.github.io/kallisto/ |
| Software, Algorithm | HISAT2 | PMID: 25751142 | RRID:SCR_015530 | https://ccb.jhu.edu/software/hisat2/index.shtml |

*Continued on next page*

*Continued*

| Reagent type or resource | Designation | Source of reference | Identifiers | Additional information |
|---|---|---|---|---|
| Software, Algorithm | Samtools | PMID: 19505943 | RRID:SCR_002105 | http://samtools.sourceforge.net/ |
| Software, Algorithm | Jalview | PMID: 19151095 | RRID:SCR_006459 | http://www.jalview.org/ |
| Software, Algorithm | MAFFT | PMID: 23329690 | RRID:SCR_011811 | https://mafft.cbrc.jp/alignment/software/ |
| Software, Algorithm | IQ-Tree | PMID: 25371430 | RRID:SCR_017254 | http://www.iqtree.org |
| Software, Algorithm | ITOL | PMID: 27095192 | | https://itol.embl.de/ |
| Software, Algorithm | PAL2NAL | PMID: 16845082 | | |

## Strains

The following strains were used in this study:

### Near isogenic lines (NILs)

CX12311 (N2*): *kyIR1(V, CB4856 > N2), qgIR1(X, CB4856 > N2)*,
PTM66 (NIL$_{(nurf-1,LSJ2>N2*)}$): *kyIR87(II, LSJ2 > N2); kyIR1(V, CB4856 > N2), qgIR1(X, CB4856 > N2)*
CRISPR-generated allelic replacement lines (ARLs)
PTM88 (ARL$_{del, LSJ2>N2}$): *kyIR1(V, CB4856 > N2); qgIR1(X, CB4856 > N2); nurf-1(kah3)II; spe-9(kah132)I*
PTM416 (ARL$_{intron,LSJ2>N2}$): *nurf-1(kah127)II*
PTM417: *kyIR1(V, CB4856 > N2); qgIR1(X, CB4856 > N2); nurf-1(kah3)II*

### CRISPR-generated barcoded strains

PTM229: *dpy-10(kah82)II*
PTM288: *kyIR1(V, CB4856 > N2); qgIR1(X, CB4856 > N2); dpy-10(kah82)II*

### CRISPR-generated epitope-tagged strain

PTM420 (HA-FLAG): *nurf-1(kah124,kah133)II,*

### CRISPR-generated STOP codons replacement lines

PTM98 (exon23): *nurf-1(kah11)II*
PTM203 (exon26): *nurf-1(kah68)II*
PTM316 (exon 1): *nurf-1(kah90)II/oxTi924 II*
PTM317 (exon 2): *nurf-1(kah91)II/oxTi924 II*
PTM319 (exon 15): *nurf-1(kah93)II/oxTi924 II*
PTM322 (exon 18): *nurf-1(kah96)II/oxTi924 II*
PTM325 (exon 19): *nurf-1(kah99)II/oxTi924 II*
PTM332 (exon 2): *nurf-1(kah106) II/oxTi924 II*
PTM487 (exon 7): *nurf-1(kah142) II/oxTi721 II*

### CRISPR-generated domain replacement lines

PTM113 (PHD1): *nurf-1(kah16)II,*
PTM116 (PHD2): *nurf-1(kah19)II,*
PTM117 (PHD2): *nurf-1(kah20)II,*
PTM118 (Bromodomain): *nurf-1(kah21)II,*
PTM167 (Bromodomain): *nurf-1(kah32)II,*

PTM170 (double PHD): *nurf-1(kah19,kah36)II*,
PTM189 (three domains): *nurf-1(kah19,kah36,kah54)II*,
PTM211 (double PHD): *nurf-1(kah66,kah73)II*

## MosSCI transgenic strains

PTM371: *nurf-1(kah93) II/oxTi721 II; kahSi7*,
PTM372: *nurf-1(kah96) II/oxTi721 II; kahSi7*,
PTM373: *nurf-1(kah99) II/oxTi721 II; kahSi7*,
PTM376: *nurf-1(n4295) II; kahSi7*,
PTM517: *kyIR1 (V, CB4856 > N2); qgIR1 (X, CB4856 > N2); nurf-1(kah3) II; kahSi7*

## CRISPR-generated deletion strains:

PTM512 (23$^{rd}$ exon deletion): *nurf-1(kah149) II*
PTM489 (HA-FLAG + 23$^{rd}$ exon deletion): *nurf-1(kah124,kah133,kah144)II*

## Other double mutants:

PTM354: *nurf-1(n4295, kah113) II/oxTi924 II*

## **Strain construction**

## Previously described strains

CX12311, PTM66, and PTM88 were all previously described (**McGrath et al., 2011**; **Large et al., 2016**).

## CRISPR-generated allelic replacement lines (ARLs)

We used the coCRISPR protocol to generate all CRISPR-edited lines using single-strand oligonucleotides to make precise edits (**Arribere et al., 2014**; **Paix et al., 2015**).

Resequencing of the PTM88 strain identified a number of background mutations, including an A to G missense SNV that is predicted to change an asparagine to an aspartic acid which we named *kah132*. The flanking sequence of this mutation is 5'-cgacaatgac[a]atcgccaggg-3'. We backcrossed out this *spe-9(kah132)* mutation, along with additional background mutations, to create PTM417.

To create PTM416, we designed a number of guide RNAs nearby the intron SNV. However, we were unable to identify editing events using these guide RNAs, putatively due to the high usage of As and Ts.

We turned to a two-step strategy to create the edit, first creating a deletion of the 2$^{nd}$ intron along with flanking exon regions using guide RNAs with high predicted efficiency. We created the following constructs driving the following sgRNAs:

5'- TCGATAATTATCCGTTTGT(GGG) −3',
5'- TTGCATCATATCCCACAAA(CGG) - 3',
5'- ACGGTAGCTCATGAAGAGA(AGG) −3' and 5'- TTCCGACGAATATAAGAAA(CGG) −3'

We also ordered an oligonucleotide repair:

5'-GTCTGTTAGAGATGCTATTAATGTCGATAATTATCgctaccataggcaccacgagcgagATTCG
TCGGAATTTAAGAAACTTGTGAATAATGTT −3'

We injected 50 ng/µl of P$_{eft-3}$::Cas9, 25 ng/µl of *dpy-10* sgRNA, 500 nM *dpy-10(cn64)* repair oligo, 10 ng/µl of each of the *nurf-1* sgRNAs listed above, and 500 nM of the repair oligonucleotide into CX12311 animals.

Jackpot broods were identified and roller animals were genotyped using the following primers along with the BanI restriction enzyme:

5'- GCAGGCCGGCCTTCGCGCCTGGGTAATACC −3' and
5'- CGGCAGTTTTCGTCGTTCTG −3'

A single heterozygote worm was identified. Wild-type heterozygote progeny were identified (to remove the linked *dpy-10* mutation) and this mutation was balanced (homozygous animals were sterile) with an integrated GFP marker near the *nurf-1* gene (*oxTi924*). This strains was frozen with the

following genotype: PTM366 *nurf-1(kah125)/oxTi924* II; *kyIR1 (V, CB4856 > N2); qgIR1 (X, CB4856 > N2) X*.

For the second step, we crossed PTM366 to PTM66 animals and selected non-fluorescing animals to create *nurf-1(kah125)/kyIR87(II, LSJ2 > N2); kyIR1 (V, CB4856 > N2); qgIR1 (X, CB4856 > N2) X* compound heterozygote animals. We used the following sgRNAs to specifically target the *nurf-1 (kah125)* homologous chromosome:

5'- ATctcgctcgtggtgccta(tgg) −3' and 5'- TTCCGACGAATctcgctcg(tgg) −3'

The 2^nd homologous chromosome, containing the *kyIR87* introgression was used as a repair construct. We injected 50 ng/µl P*eft-3*::Cas9, 10 ng/µl *dpy-10* sgRNA, 500 nM *dpy-10(cn64)* repair oligo and 25 ng/µl of each *nurf-1* sgRNA. Roller animals were then PCR genotyped to screen for animals that were homozygous for the LSJ2 allele at the intron and heterozygote for the 60 bp deletion.

After screening, the target genotype was made homozygous. This strain was named PTM410 *kyIR1 (V, CB4856 > N2); qgIR1 (X, CB4856 > N2); nurf-1(kah127)II*. PTM416 was created by backcrossing the PTM410 strain to the N2 background using an RFP fluorescent *nurf-1* balancer (*oxTi721*) strain for four generations. We genotyped the *npr-1* and *glb-5* sites to verify that PTM416 did not carry the introgressions surrounding these genes.

## CRISPR-generated isotope-tagged lines

To create the PTM420 epitope-tagged strain the following guide RNA and repair oligo was used to first add an HA epitope tag into the 16^th exon:

5'-TGGCACTTGCTCAGTTGTGG-3'
5'-TTTTGTCAAATTTGGAGCCGTTTGGGGAACCTCTAggcgtagtcggggacgtcgtatggg-tatcctcctcctcctcctcccTGcTGtTCgTCTGGgACcTGCTCgGTTGTaGTaGAAACTGCGAAACCAGTCGCG TCATCAGGCATGTC-3'

The following injection mix was used: 50 ng/µl P*eft-3*::Cas9, 10 ng/µl *dpy-10* sgRNA, 500 nM *dpy-10(cn64)* repair oligo, 25 ng/µl of sgRNA, and 500 nM repair oligonucleotide.

We next added a 3xFLAG tag to the C-terminal of *nurf-1* gene using purified Cas9 protein (IDT, Catalog #1074181) and in vitro synthesized RNAs (Synthego) using a modified protocol (*Prior et al., 2017*). The injection mix was prepared as follows: 2 µM *dpy-10* sgRNA (RNA scaffold 5'- GCUACCA UAGGCACCACGAG −3' + tracrRNA) and 4 µM of two sgRNAs that targeted this region (RNA scaffold: 5'- CUCAUAAGUUCGCAUCCAG −3'+ tracrRNA, 5'- UUCGGAUCAGCUGUUGCCAC −3'+ tracrRNA) were mixed and incubated in a thermocycler at 95°C for five minutes, then 2.5 µg/ul Cas9 protein was added and incubated at room temperature for five minutes. Finally, 0.2 µM *dpy-10* repair oligo and 0.5 µM FLAG repair oligo were added to mix and incubate at room temperature for 60 min. This mix was injected into the HA-tagged strain to create the double epitope tagged line.

## CRISPR-generated STOP codon replacement lines, PHD/bromodomain replacement lines, and deletion lines

The following injection mix was used to create each of these strains: 50 ng/µl P*eft-3*::Cas9, 10 ng/µl *dpy-10* sgRNA, 500 nM *dpy-10(cn64)* repair oligo, 25 ng/µl of sgRNA, and 500 nM repair oligonucleotide. For each strain/allele, each of the specific sgRNAs and repair oligos used to construct it are listed in *Supplementary file 3*. To facilitate the genotyping process, some of the repair oligos for STOP codon replacement sites contain restriction sites that will alter some of the amino acids, exact changes are listed in Supplementary file 4. In *C. elegans* nomenclature, Identical edits must be given different allele names if they were isolated independently.

For mutants that were sterile (or lead to sterility), we balanced these mutations using a GFP (*oxTi924*) or mCherry (*oxTi721*) integrated marker near *nurf-1*.

## MosSCI transgenic strains

MosSCI strain construction was done following standard protocol from Frøkjær-Jensen et. Al (*Frøk-jær-Jensen, 2015*). Injection mix was prepared as following: 38 ng/ul pCFJ601 (Mos1 transposase), 30 ng/ul pCFJ151 - P*nurf-1.d*::*nurf-1.d-SL2-GFP* (insertion vector with homologous arms), 2.5 ng/ul pCFJ90 (P*myo-2*::mCherry), 5 ng/ul pCFJ104). This was injected into EG6699 uncoordinated animals. Three injected animals were placed on a single plate at 30°C to facilitate starvation. After 5 days, coordinated animals with GFP fluorescence and no red fluorescence were singled to new NGM

plates and allowed to proliferate. Their progenies were singled and a single homozygote without uncoordinated offspring was maintained. This homozygote was then backcrossed to N2 for four generations to remove *unc-119(ed3) III* to create the PTM337 strain containing the integrated rescue construct. This strain was then crossed to a variety of *nurf-1* alleles using standard protocols.

## Cell culture

The following human cancer cell lines were used: Colo-205 (colorectal), MCF-7 and MDA-MB-231 (breast), and HeLa (cervix) were obtained from the American Type Culture Collection (Rockville, MD); A549 (lung) was kindly provided by G. Roncador, CNIO. Cells were authenticated using STR profiling, tested for mycoplasma contamination and negative. Cells were cultured in DMEM (Sigma-Aldrich) supplemented with 10% FBS (HyClone, Logan, UT, USA), except for A549 which were cultured in RPMI (Sigma-Aldrich) supplemented with 10% FBS and sodium pyruvate (Thermo Scientific).

## Molecular biology

All sgRNAs were constructed using NEB Q5 site directed mutagenesis kit (E0554) using primers

5'- [unique sgRNA protospacer sequence] + GTTTTAGAGCTAGAAATAGCAAGT −3' and

5'- CAAGACATCTCGCAATAGG −3' to modify a vector backbone containing a subclone of pDD163 containing the U6 promoter to drive sgRNAs in germline[1].

To create the pCFJ151 - P*nurf-1.d::nurf-1.d-sl2-GFP* plasmid, a *nurf-1.d* cDNA was isolated from reverse transcribed RNA using primers containing NheI restriction sites. This PCR product was then digested and ligated to a pSM vector. A 2890 bp long promoter region immediately upstream of the *nurf-1.d* isoform was amplified with a forward primer including FseI and a reverse primer including AscI restriction sites. This PCR product was then digested and ligated into the vector constructed in step 1. Third, an SL2-GFP sequence from was cut and ligated into the new vector using KpnI and SpeI restriction sites. Finally, this entire sequence containing the promoter, cDNA and sl2:: GFP sequence was inserted into the pCFJ151 vector using NEB Q5 site directed mutagenesis kit.

## Nematode growth conditions

The animals were cultured on 6 cm standard nematode growth medium (NGM) plates containing 2% agar seeded with 200 μl of an overnight culture of the *E. coli* strain OP50. Growth temperature was controlled using a 20°C incubator. Strains were grown for at least three generations without starvation before any experiments was conducted.

## *nurf-1* conserved regions

The predicted protein sequence for the NURF-1.A protein isoform was BLAST-searched against human or *Drosophila melanogaster* protein databases using NCBI blastp (*McGinnis and Madden, 2004*). Regions with alignment scores above 50 were annotated as homologous regions. These homologous regions were further verified through multiple sequence alignmentwith Clustal Omega program (*Chojnacki et al., 2017*).

## Competition experiment

Competition experiments were performed as described previously (*Zhao et al., 2018*).

## RNA-seq analysis

### RNA-seq samples for comparing the effect of the nurf-1 intron SNV

N2 and PTM416 worms were synchronized using a 3 hr hatch-off. Worms were observed every hour after 46 hr until the majority were in the L4 stage (which occurred at 48 hr). Four hours later, worms were collected and kept frozen in −80°C freezer until RNA extraction for the 52 hr timepoint. Eight hours later, young adult animals were collected and kept frozen in the −80°C freezer until RNA extraction for the 60 hr timepoint.

### RNA-seq samples for comparing effect of the two derived nurf-1 mutations

CX12311, PTM66, PTM88, LSJ2 L4 hermaphrodites were picked to fresh NGM agar plates. Their adult progeny were bleached using alkaline-bleach solution to isolate eggs for synchronization. The eggs were washed with M9 buffer for three times and placed on a tube roller overnight. About 400

hatched L1 animals were placed on NGM agar plates and incubated at 20°C until they reach young adulthood, as determined by when eggs were observed on assay plates. These worms were then harvested, washed 3 times with M9 buffer, and frozen in a −80°C freezer for later processing.

## RNA-seq samples for heat shock

N2 and PTM416 worms were synchronized using a 3 hr hatch-off. Eggs were cultured at 20°C until they reached L4 stage. Heat shock assay plates were then wrapped with parafilm and placed in a water bath pre-heated to 34°C for 2 hr or 4 hr. Worms were either collected right after heat shock or after 30 min at 20°C for the recovery group.

For each of the above experiments, RNA was isolated using Trizol. The RNA libraries were prepared using an NEBNext Ultra II Directional RNA Library Prep Kit (E7760S) following its standard protocol. The libraries were sequenced by an Illumina NextSeq 500. The reads were aligned by HISAT2 using default parameters for pair-end sequencing (*Kim et al., 2015*). These aligned reads were then visualized in IGV browser (*Robinson et al., 2011*) to examine *nurf-1* splice junction track (as shown in *Figure 2—figure supplement 1*). Transcript abundance was calculated using feature-Count and then used as inputs for the SARTools. SARTools use edgeR for normalization and gene-level differential analysis (*Varet et al., 2016*) and output the multidimensional scaling plot for each transcriptome analysis project. Differentially expressed genes were determined for comparisons have adjusted p-value<0.05. Genes upregulated and downregulated are plotted separately for the tissue and stage analysis. Each gene was normalized by dividing the sum of its expression level across all stages and this normalized table was used for hierarchical clustering analysis. Sequencing reads were uploaded to the SRA under PRJNA526473.

Kallisto was used to quantify abundances of *nurf-1* transcripts (*Bray et al., 2016*). We first created our own reference transcriptome by modifying the transcripts in Wormbase published reference transcriptome to restrict our analysis to the *nurf-1.a, nurf-1.b, nurf-1.d, nurf-1.f* and *nurf-1.q* isoforms. Alternative splicing sites in the 10th, 16th, and 21st exons were also removed from this reference database to ensure they were consistent between all isoforms. We used wildtype L2 RNA-seq data from Brunquell et. al to quantify wildtype *nurf-1* abundance (*Brunquell et al., 2016*) and extracted tpm(transcripts per million) data from Kallisto output abundance table. We used RNA-seq data from PRJNA311958 and PRJNA321853 (*Brunquell et al., 2016*) (*Li et al., 2016*) to quantify the heat shock response of *nurf-1* isoforms in *Figure 4—figure supplement 2B*.

## Western blot

4 N2 and PTM420 gravid hermaphrodites were picked to fresh 5.5 cm NGM agar plates. Worms were collected just prior to starvation using M9 buffer and stored at −80°C until protein extraction. At least 4 plates of worms were used for each protein isolation. Worms were condensed by centrifugation and 2x sample buffer (100 mM Tris-HCl pH 6.8M, 200 mM dithiothreitol, 4% SDS, 0.2% Bromophenol Blue, 20% glycerol) was added in 1:1 w/v ratio. 1 μl of 500 mM EDTA and 1 μl of Halt protease inhibitor cocktail (100x) (Catalog number: 78430) were added for every 100 ng of worm sample. The protein sample was vortexed for 90 s and incubated on ice for about 1 min. Samples were then sonicated in a Bransonic 0.5 gallon ultrasonic bath filled with hot water > 80°C for 10 min and immediately placed on ice for 2 min. We then boiled the samples for 5 min and placed on ice to cool down. The sample was centrifuged at 12,000 rpm for 5 min and the supernatant was transferred to new tubes.

All samples were loaded on 5% SDS-PAGE gel at 3 μl, 5 μl and 7 μl volumes followed by Coomassie blue staining and washing steps. Gels were then dried using DryEase Mini-Gel Drying System (Invitrogen, Catalog number: NI2387). These gels were used to normalize protein loading volume for different samples.

Each sample was loaded onto a freshly made 6% or 10% SDS-PAGE gel and run at 25 mA. Gel samples were then transferred in 10 mM CAPS pH 10.5 buffer at 20 V and 20 mA for 17 hr to a PVDF membrane. Protein products with HA tag were detected using 1:500 anti-HA antibody (Life Technologies, Catalog number: 326700), NURF-1.D isoform with FLAG tag was detected using 1:1000 PIERCE ANTI-DYKDDDDK antibody (Life Technologies, Catalog number: MA191878) and NURF-1.F isoform with FLAG tag was detected using 1:1000 Millipore ANTI-FLAG antibody (Millipore Sigma, Catalog number: F3165).

For western blots of cancer cell lines, cells were lysed in 1% NP-40 buffer supplemented with protease and phosphatase inhibitors. Following sonication, clearing by centrifugation, and protein quantification, samples (100 µg) were subjected to electrophoresis in NuPAGE 3–8% Tris-acetate precast polyacrylamide gels (Thermo Scientific). Samples were run under reducing conditions and then transferred to nitrocellulose membranes, which were blocked with TBST, 5% skim milk. Membranes were incubated with primary antibodies detecting the following proteins: BPTF (NB100-41418, Novus Biologicals) (1:1,000) and Vinculin (V9131-2ML, Sigma-Aldrich) (1:10,000). This was followed by incubation with horseradish peroxidase-conjugated secondary antibodies (Dako, Glostrup, Denmark) (1:10,000). Reactions were detected using an ECL detection system and Bio-Rad ChemiDoc MP Imaging System (Hercules, CA, USA).

## Egg-laying analysis

Egg laying assays were performed as previously described (*Large et al., 2016*). All egg-laying assays were carried out at 20℃ using standard 3 cm NGM plates seeded with the OP50 strain of *Escherichia coli*. OP50 were prepared freshly by streaking a glycerol stock of OP50 on an LB plate and letting grow at 37℃ overnight. A single colony was then picked to 5 ml fresh LB and cultured overnight in a shaking incubator at 200 rpm. 1 ml of the overnight culture was used to inoculate 200 ml of LB for 4–6 hr of growth at 37℃ with shaking. The 200 ml OP50 culture was concentrated via centrifugation to an OD600 of 2.0 and this culture was used for seeding experimental plates with 50 µl aliquots. All experimental plates were prepared the week of the assay and left at 22.5℃ 18–24 hr following seeding. Plates were then placed at 4℃ until the day of the assay and warmed to 20℃ for 12 hr before each time point.

For strains that have severe reduced fertility when homozygous, one L4 nematode was transferred to the 50 µl experimental plate. The number of eggs laid were measured every 12 or 24 hr, and eggs laid per hour was calculated by dividing the time range and number of animals left on each plate at each timepoint. At least 10 replicates were assayed for each strain.

For other strains, six fourth larval stage (L4) nematode was transferred to the 50 µl experimental plate. The number of eggs laid were measured every 12 or 24 hr, and eggs laid per hour was calculated by dividing the time range and number of animals left on each plate at each timepoint. Six replicates were assayed for each strain.

Fecundity was calculated by summing up all eggs laid for each worm.

## Analysis of growth rate using body sizes

For strains with mutations in PHD or bromodomains, growth analysis were performed as previously described (*Large et al., 2016*). For other strains, video recordings were analyzed similarly, with the exception that each animal was registered between each video frame and used to calculate an average area for each individual worm. For strains that were balanced with fluorescent markers, only non-fluorescent worms were picked for video tracking.

## Sperm and oocyte counting analysis

4 N2, PTM332, PTM319 and PTM332 gravid hermaphrodites were picked to fresh 5.5 cm NGM agar plates. After 3 days, 20–30 non-fluorescent L4 worms were picked to a new NGM plate and let grow at 20℃ for 12 hr. Worms were then picked to a drop of M9 buffer on a Fisher Superfrost Plus slide (22-037-246). Fixation was done through applying 95% ethanol for three times. A drop of Vector Laboratories Vectashield Mounting Medium with DAPI (H-1500) was added and a coverslip was applied and sealed with nail polish. Z-stack images were captured through a moving-stage Olympus IX73 microscope under 40x objective. Oocytes were counted while imaging and sperm number was measured manually by analyzing z-stack images on ImageJ through the CellCounter plugin.

## Genomic and transcriptomic analysis of *nurf-1* in additional *Caenorhabditis* species

To identify *nurf-1* orthologs, we used homology information included in www.wormbase.org or by BLAST-searching *C. elegans* protein sequences against protein data provided by the *Caenorhabditis* genome project (http://blast.caenorhabditis.org). Genomic regions that contain the identified *nurf-1* orthologs and related gff3 annotation data were downloaded from download.caenorhabditis.org or

the WormBase public FTP site (data from *Stein et al., 2003*) (*Mortazavi et al., 2010*; *Fierst et al., 2015*; *Slos et al., 2017*; *Kanzaki et al., 2018*; *Yin et al., 2018*; *Lamelza et al., 2019*). Species with public RNA-seq data were identified in the SRA database. These reads were downloaded and aligned to corresponding *nurf-1* DNA reference sequence for each species using HISAT2 and further manipulated using SAMTOOLS (*Li et al., 2009*; *Kim et al., 2015*). Gene annotations were manually corrected by inspecting the RNA-seq predicted intron sequences and used to generate Sashimi plots using the IGV browser (*Robinson et al., 2011*; *Katz et al., 2015*). The Sashimi Plot parameter Junction Coverage Min was adjusted for each species to best visualize the exon-exon junctions based upon coverage data. To identify the duplicated region for the NURF-1.B and NURF-1.D isoforms, we blasted each B isoform against a database of the D isoforms, and vice-versa. The homologous regions for each protein were refined using a multiple sequence alignment of NURF-1.B and NURF-1.D proteins using Jalview (*Waterhouse et al., 2009*). For some of the species that we were unable to resolve the full *nurf-1* region (due to missing sequence for part of the region), we were able to identify the duplicated region and included this in the phylogenetic analysis.

## Phylogenetic analysis

We aligned the protein sequences of the duplicated region from the *nurf-1* loci of 21 *Caenorhabditis* species using MAFFT (*Katoh and Standley, 2013*). We also aligned the protein sequences for regions outside the duplicated region. Maximum likelihood trees were estimated for each alignment along with 1000 ultrafast bootstraps (*Hoang et al., 2018*) using IQ-TREE (*Nguyen et al., 2015*), allowing the best-fitting substitution model to be automatically selected (*Kalyaanamoorthy et al., 2017*). We noted that the resulting topology recovered for the duplicated region was incongruent with the species tree, likely due to limited phylogenetic signal in the short alignment (*Figure 6—figure supplement 6*). To address this, we instead assessed the levels of support for alternative phylogenetic hypothesis surrounding the number and timing of duplication events that we congruent with the species tree. Log-likelihoods were calculated for each topology and an approximately unbiased (AU) test (*Shimodaira, 2002*) was performed using IQ-TREE. Newick trees were visualized using the iTOL web server (*Letunic and Bork, 2016*).

For three pairs of closely-related sister taxa (*C. briggsae/C. nigoni, C. latens/C. remanei,* and *C. afra/C. sulstoni*), we aligned the protein sequences of both *nurf-1–1 (nurf-1.b)* and *nurf1-2 (nurf-1.d)* using MAFFT and converted the resulting alignments to nucleotide alignments using PAL2NAL (*Suyama et al., 2006*). We calculated the dN/dS ratio (Ka/Ks) separately for the duplicated and non-duplicated portions of each alignment using the dnds Python module (available at: https://github.com/adelq/dnds).

## Statistics

Sample size was calculated by following replicate numbers using previously published assays. Each data point was considered a biological replicate. Animals for each replicate were grown independently for at least three generations. Significant differences between two means were determined using two-tailed unpaired t-test. To correct for multiple comparison, we used the Tukey multiple comparison test.

## Proteomics

MCF-7 whole cell extracts were obtained by lysis in either NP-40 (see above) or Laemmli buffer, in both cases supplemented with protease inhibitors and loaded in NuPAGE 3–8% Tris-acetate precast polyacrylamide gels (75 µg of protein per well). Gels were cut into two slices for western blotting and Coomassie staining. Gels bands running at the mobility of BPTF signals detected by western were digested with trypsin as previously described (*Shevchenko et al., 2006*). Briefly, gel bands were cut into 1 mm$^2$ cubes and de-stained with 50 mM ammonium bicarbonate (ABC) solution. Then proteins were reduced with 15 mM TCEP and alkylated with 30 mM CAA at 45°C, for 45 min in the dark. Proteins were digested with 200 ng of Trypsin (Promega) overnight at 37°C in 50 mM ABC. Resulting peptides were desalted using homemade reversed phase micro-columns containing C18 Empore disks (3M) at the bottom of the tip. Samples were dried down using a Speed-Vac and dissolved in 22 µL of loading buffer (0.2% formic acid) prior LC-MS/MS analysis.

LC-MS/MS was performed by coupling an Ultimate 3000 RSLCnano System (Dionex) with a Q-Exactive Plus mass spectrometer (Thermo Scientific). Peptides were loaded into a trap column (Acclaim PepMap 100; 100 μm × 2 cm; Thermo Scientific) over 3 min at a flow rate of 10 μl/min in 0.1% formic acid (FA). Then peptides were transferred to an analytical column (PepMap rapid separation liquid chromatography C18; 2 μm, 75 μm × 50 cm; Thermo Scientific) and separated using a 90 min effective linear gradient (buffer A: 0.1% FA; buffer B: 100% acetonitrile, 0.1% FA) at a flow rate of 250 nl/min. The gradient used was as follows: 0–5 min 4% B, 5–7.5 min 6% B, 7.5–60 min 17.5% B, 60–72.5 min 21.5% B, 72.5–80 min 25% B, 80–94 min 42.5% B, 94–94.1 min 98% B, 94.1–99.9 min 98% B, 99.9–100 min 4% B and 100–104.5 min 4% B. The peptides were electrosprayed (2.1 keV) into the mass spectrometer through a heated capillary at 300°C and an S-Lens radio frequency (RF) level of 50%. The mass spectrometer was operated in a data-dependent mode, with an automatic switch between the MS and MS/MS scans using a top 15 method (minimum automatic gain control target, 3E3) and a dynamic exclusion time of 26 s. MS (350–1,400 m/z), and MS/MS spectra were acquired with a resolution of 70,000 and 17,500 full width at half maximum (FWHM; 200 m/z), respectively. Peptides were isolated using a 2 Thompson unit (Th) window and fragmented using higher-energy collisional dissociation at 27% normalized collision energy. The ion target values were 3E6 for MS (25 ms maximum injection time) and 1E5 for MS/MS (45 ms maximum injection time).

Raw files were processed with MaxQuant (v 1.6.2.6) using the standard settings against a human protein database (UniProtKB/Swiss-Prot, 20,373 sequences) including all annotated BPTF isoforms deposited in TrEMBL and supplemented with contaminants. Carbamidomethylation of cysteines was set as a fixed modification whereas oxidation of methionines and protein N-term acetylation were set as variable modifications. Minimal peptide length was set to seven amino acids and a maximum of two tryptic missed-cleavages were allowed. Results were filtered at 0.01 FDR (peptide and protein level).

## Acknowledgements

We thank the *Caenorhabditis* Genetics Center, which is funded by NIH Office of Research Infrastructure Programs (P40 OD010440), for strains, and WormBase for information. We are grateful to Rachael Workman and Winston Timp for sharing Oxford Nanopore reads of *nurf-1* prior to publication. We thank Matthew Rockman, Luke Noble, Janna Fierst, Erich Schwarz, and Janet Young for access to unpublished genomic data. We thank F.X Real for support, valuable discussions, and comments on the manuscript and G Roncador and the CNIO Monoclonal Antibody Core Unit for helpful contributions. We also thank Todd Streelman, Greg Gibson, Soojin Yi, David Katz, Annalise Paaby, and members of the McGrath lab for discussions, and Annalise Paaby and Erik Andersen for comments on the manuscript. This work was supported by NIH R01GM114170 (to PTM), R01GM121688 (to R E E), and a CNIO friends/Juegaterapia grant (to IF). Work at CNIO was supported, in part, by grant RTI2018-101071-B-I00 from Ministerio de Ciencia, Innovación y Universidades. CNIO is supported by Ministerio de Ciencia, Innovación y Universidades as a Centro de Excelencia Severo Ochoa SEV-2015–0510.

## Additional information

### Funding

| Funder | Grant reference number | Author |
| --- | --- | --- |
| National Institute of General Medical Sciences | R01GM114170 | Patrick T McGrath |
| National Institute of General Medical Sciences | R01GM121688 | Ronald E Ellis |

The funders had no role in study design, data collection and interpretation, or the decision to submit the work for publication.

## Author contributions
Wen Xu, Conceptualization, Resources, Data curation, Formal analysis, Validation, Investigation, Visualization, Methodology, Writing—original draft, Writing—review and editing; Lijiang Long, Resources, Data curation, Formal analysis, Investigation, Visualization, Writing—review and editing; Yuehui Zhao, Formal analysis, Investigation, Writing—review and editing; Lewis Stevens, Formal analysis, Investigation, Visualization, Methodology, Writing—review and editing; Irene Felipe, Data curation, Formal analysis, Validation, Investigation, Visualization, Methodology; Javier Munoz, Data curation, Formal analysis, Validation, Investigation; Ronald E Ellis, Conceptualization, Formal analysis, Funding acquisition, Investigation, Writing—review and editing; Patrick T McGrath, Conceptualization, Formal analysis, Supervision, Funding acquisition, Validation, Investigation, Visualization, Methodology, Writing—original draft, Project administration, Writing—review and editing

## Author ORCIDs
Wen Xu http://orcid.org/0000-0003-2085-7223
Lijiang Long https://orcid.org/0000-0002-9897-5900
Yuehui Zhao https://orcid.org/0000-0002-9496-0023
Lewis Stevens http://orcid.org/0000-0002-6075-8273
Javier Munoz https://orcid.org/0000-0003-3288-3496
Patrick T McGrath https://orcid.org/0000-0002-1598-3746

## Decision letter and Author response
Decision letter https://doi.org/10.7554/eLife.48119.049
Author response https://doi.org/10.7554/eLife.48119.050

# Additional files
## Supplementary files
• Supplementary file 1. RNA-seq counts for each gene.
DOI: https://doi.org/10.7554/eLife.48119.038

• Supplementary file 2. GO Category analysis for intron SNV regulon.
DOI: https://doi.org/10.7554/eLife.48119.039

• Supplementary file 3. Guide RNAs for CRISPR-Cas9 genome edits.
DOI: https://doi.org/10.7554/eLife.48119.040

• Transparent reporting form
DOI: https://doi.org/10.7554/eLife.48119.041

## Data availability
Sequencing reads were uploaded to the SRA under PRJNA526473.

The following dataset was generated:

| Author(s) | Year | Dataset title | Dataset URL | Database and Identifier |
|---|---|---|---|---|
| Xu W, Long L, McGrath P | 2019 | RNAseq of *C. elegans* under different genetic background and heat shock treatment to study the roles of different isoforms of nurf-1 | https://www.ncbi.nlm.nih.gov/bioproject/PRJNA526473 | NCBI Sequence Read Archive, PRJNA526473 |

The following previously published datasets were used:

| Author(s) | Year | Dataset title | Dataset URL | Database and Identifier |
|---|---|---|---|---|
| Jian Li, Laetitia Chauve, Grace Phelps, Renée M Brielmann, Richard I Morimoto | 2016 | RNA-seq analysis in *C. elegans* larval development and heat shock | https://www.ncbi.nlm.nih.gov/bioproject/?term=PRJNA321853 | NCBI Sequence Read Archive, PRJNA321853 |

Jessica Brunquell, Stephanie Morris, Yin Lu, Feng Cheng, Sandy D Westerheide | 2016 | The genome-wide role of HSF-1 in the regulation of gene expression in *Caenorhabditis elegans* | https://www.ncbi.nlm.nih.gov/bioproject/PRJNA311958/ | NCBI Sequence Read Archive, PRJNA311958

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
