## [Decision Letter]

[Editors’ note: this article was originally rejected after discussions between the reviewers, but the authors were invited to resubmit after an appeal against the decision.]

Thank you for submitting your work entitled "Evolution of Yin and Yang isoforms of a chromatin remodeling subunit results in the creation of two genes" for consideration by *eLife*.

We regret to inform you that we cannot, at present, accept your paper for publication in *eLife*. Because your paper's scientific work is of high quality, we are willing to reconsider a thoroughly revised version. However, we can only do so if you can satisfactorily address each of the three problems discussed below.

Your article has been reviewed by three peer reviewers, including Erich M Schwarz as the Reviewing Editor and Reviewer #1, and the evaluation has been overseen by a Senior Editor. The following individual involved in review of your submission has agreed to reveal their identity: Eric Haag (Reviewer #3). Our decision has been reached after consultation between the reviewers.

The problems raised by the reviewers that necessitated this decision are as follows.

1) The domestication story is interesting, and the evidence that the first intron harbors an important variant vis-a-vis fitness in lab culture is quite strong. However, there is no mechanistic explanation for how it impacts nurf-1 function. This is a major deficiency for an *eLife* paper.

2) The authors prove that the full-length transcript of nurf-1 is not needed, and they provide evidence that one clade of *Caenorhabditis* have split the ancestral, single complex nurf-1 gene. However, there remains the question of whether nurf-1's 5'-ward and 3'-ward halves remain connected as part of one operon, with SL2 splicing to the 3'-ward half. Moreover, why the splitting of nurf-1 happened is a) not clear, and b) not connected to the domestication phenotype. It is thus a curiosity unrelated to a major biological event (thus far).

3) Although some evidence of accelerated evolution of the overlap region between Yin and Yang isoforms is provided, there is no compelling evidence that there was an "adaptive conflict" that needed to be resolved and was indeed resolved. The null hypothesis of "both work, and developmental systems drift happens" cannot be rejected. This issue is at the core of your article's claim to biological importance.

All three reviewers agreed that the paper has abundant experimental data of high quality. The problem is that these data are not convincingly linked to the explanations claimed, and the results seem to fall naturally into two distinct papers (one on domestication alleles, the other on the molecular biology and evolution of isoforms arising from the genetically complex nurf-1 locus). Unless the three points above can be addressed, this work may fare better as two shorter studies focused on these two topics.

We have attached the individual reviews by the three reviewers below.

Reviewer #1:

Using molecular biology, reverse genetics, and transcriptomics, Xu et al. make a compelling case for the evolution of two isoforms of nurf-1/BPTF in *Caenorhabditis elegans* that subdivide functions within this gene; they go on to show that a subgenus of *Caenorhabditis* species have evolved two entirely separate genes from these isoforms. Their observations provide a coherent explanation for many functions of the nurf-1 gene that had been intermittently seen over the years, but not reconciled. Their analysis of nurf-1 provides a beautiful instance of evolutionary adaptation, both in the sense of nurf-1's evolving into two functional isoforms, and in the sense that nurf-1 itself (as the authors show) mediates phenotypic tradeoffs in life history between early spermatocyte production and later oocyte production, which ultimately affects speed and output of the *C. elegans* reproductive life cycle that have different optima in different laboratory environments (plate culture for the N2 strain, versus liquid culture for the LSJ2 strain).

The experimental work described here is thorough, rigorous, and adroit.

The manuscript is very well composed. Background material is scholarly but appropriately concise. Generally, text is clean and direct. I was struck by the clarity with which the figures presented data visually. Papers such as this one that rely on genomic or transcriptomic analysis can be, in my experience, quite obtusely written or illustrated with figures that do not convey data well; Xu et al. have opposed these tendencies, for which I thank them.

I was impressed with the authors' efficient use of previously published RNA-seq data for heat shock expression, and of prepublication Nanopore long-RNA-seq data from Roach et al., 2019. By the time Xu et al. is published, I hope that the preprint Roach et al. can be cited as a published journal article.

In the text, I encountered only two points that I found difficult to decipher, a few missing data points, and one issue about nomenclature.

The first unclear point involved strain PTM417 versus strain PTM88. In Figure 1D, one of the strains shown is PTM417, but this is referred to as PTM88 in the figure legend. Reading the text carefully twice and reviewing the Materials and methods did not clarify for me which was the correct strain for Figure 1, although I suspect the correct strain is PTM417 rather than PTM88. In particular, the description of PTM417 in the Strains list (Materials and methods) is not consistent with the description of how PTM417 was constructed (subsection “CRISPR-generated allelic replacement lines (ARLs)”). I request that the authors entirely resolve this confusion.

The second unclear point involved the "Normalized Size" assay shown in Figure 3C. Two readings of the manuscript and one careful reading of the Materials and methods did not clarify, for me, what was meant by 'size' (though my guess was that it was body size). I went back to their key reference for many methods (Large et al., 2016) and concluded that the authors actually mean "Normalized body size". This should be made completely clear to the reader, both in the y-axis labeling of Figure 3C and in the legend text for Figure 3C, as well as in the Materials and methods.

One missing data point is in Table 2. This table should list a stop-codon location for kah11 in the isoform NURF-1.F, and only in that one isoform. Instead, Table 2 currently gives no stop-codon mutations for kah11 at all. In addition, Figure 3D (which is supposed to show all stop codons in their structural context) also omits kah11. These omissions from both Table 2 and Figure 3D should be corrected.

A second missing data point appears to be in the genotype table of Figure 4B. For the genotype +/kah96, this table claims that there are 0 functional copies of the isoform nurf-1.d; however, I do not see how this can be possible, given that the wild-type allele [+] should encode at least 1 functional copy of nurf-1.d. Unless there is something that I am badly misunderstanding (always possible), please correct this.

About nomenclature: the authors have dubbed the NURF-1.B isoform "Yin" and the NURF-1.D isoform "Yang". I assume that this is because of their order of expression (NURF-1.B is expressed before NURF-1.D), so that calling them "Yin and Yang" follows their times of activity. However, I found the nomenclature confusing because Yang is associated with stereotypically masculine traits and entities, whereas Yin is associated with feminine ones. Yet, the authors' current nomenclature has Yin assigned to the isoform promoting spermatogenesis (NURF-1.B), while Yang is assigned to the isoform promoting oogenesis (NURF-1.D). Since I assume that the authors would like to make it *easy* for people to properly remember which isoform does what, I would strongly encourage them to switch their nomenclature (so that NURF-1.B gets called Yang, and NURF-1.D gets called Yin).

Reviewer #2:

Here, Xu et al. report several splice isoforms of the *C. elegans* nurf-1 gene and show that they have different, possibly opposite, effects on certain aspects of gametogenesis. This has implications for fitness. The authors also found that the complex nurf-1 locus has undergone a partial duplication in several close relatives of *C. elegans*, thereby segregating distinct functions of nurf-1 into two separate genes.

1) The finding that gene expression profiles of N2 and ARL strains differ considerably at 52h, but apparently not at 60h is (Figure 1F) seems quite interesting. Surprisingly, it was not explored further. At least it should be commented on in a more elaborate way.

2) In the second paragraph of the subsection “The B and D isoforms have opposite effects on cell fate during gametogenesis”, the authors state that the observed reduction in the number of sperm is due to earlier sperm-to-oocyte switch. This is plausible, but other causes are also possible and I would encourage the authors to provide direct experimental support for this claim.

3) Are defects in sperm production of certain alleles of nurf-1 alleles (e.g. kah106) unique to hermaphrodites or are they seen in males as well? Conversely, are oocyte production defects of kah93 an issue in mutants that only produce oocytes? Addressing these questions could help to support the notion that different isoforms contribute to different aspects of the tradeoff between sperm and oocyte production. Are the observed defects in the numbers of gametes due to erroneous timing of the switch or some other problem? Can anything be said about the functions of different isoforms in gonochoristic species in which sperm vs. egg conflict is not a concern?

4) The concluding sentence of the Results section claims that the rate of amino acid replacements has accelerated in the duplicated exons. I did not find a formal test supporting this assertion.

5) Are the authors aware of this paper – Hughes, 1994?

Reviewer #3:

In this paper, Xu et al. present a meticulous examination of the nurf-1 locus of *C. elegans*. The study presents several interesting findings:

1) New evidence (beyond published studies) is presented indicating that nurf-1 has been a target of selection during domestication of laboratory strains, including the N2 and LSJ2 strains. In particular, the authors make a strong case for a major role of an intronic single nucleotide variant (SNV) in mediating adaptation to the NGM plate culture in the N2 lineage.

2) The authors then shift to a detailed characterization of the various nurf-1 transcripts and their necessity for growth and sustained fertility. This is most impressively supported by a battery of engineered stop codons. The results strongly suggest that the nurf-1.b and nurf-1.d transcripts, which overlap slightly, are the key effectors of nurf-1 function, and that the full-length nurf-1.a transcript is likely dispensable.

3) While isoform-restricted stop codons that reduce or eliminate nurf-1.b and nurf-1.d function both reduce self-fertility, they do so via opposite effects. nurf-1.b appears to be necessary to support robust spermatogenesis (i.e. it's loss leads to a partial Fog), while nurf-1.d has a role in promoting the switch from spermatogenesis to oogenesis (i.e. it is a partial Mog).

4) In the clade of Caenorhabiditis species that includes *C. brenneri*, the partially overlapping transcripts have been completely separated via a lineage-specific duplication of the exons that were historically shared.

Overall, this paper is a genetics tour de force. I do have a few suggestions, that if addressed, would tighten up the story:

1) While there appears to be a surprisingly major effect of the intronic SNV on both global gene expression and fitness, nothing is said about how this change in a homopolymeric run alters nurf-1's own gene expression. One might expect this to impact nurf-1.b transcription or splicing, and this could, in turn, alter NURF-1.B levels. Can the authors provide any data about the functional impact of the SNV on nurf-1? For example, an isoform frequency chart like that for Figure 2C, but comparing PTM228 and the ARL(intron, LSJ2>N2), would be very informative as to the mechanism by which the intron SNV impact phenotype.

2) We are told that "brood size of *C. elegans* hermaphrodites is an important trait for evolutionary fitness in laboratory conditions." Indeeed it is, but the timing also matters, with early progeny much more valuable than late progeny. Looking at the reproductive schedules in Figure 1—figure supplement 1 (and also in the Large et al., 2016 paper), there is a real shift in timing. Can the authors model, or at least speculate on, the expected impact of this?

3) In the species that have a fully separated nurf-1.1 and nurf-1.2 genes, there is very little space between the two. Have the authors looked to see if the 5' end of the transcript from the downstream gene is spliced to SL2? If so, that would indicate these have formed an operon.

---

## [Author Response]

[Editors’ note: the author responses to the first round of peer review follow.]The problems raised by the reviewers that necessitated this decision are as follows.1) The domestication story is interesting, and the evidence that the first intron harbors an important variant vis-a-vis fitness in lab culture is quite strong. However, there is no mechanistic explanation for how it impacts nurf-1 function. This is a major deficiency for an eLife paper.

To address mechanism of the intron SNV, we include analysis of nurf-1 isoform expression using RNA-seq that was already included in the paper (Figure 2—figure supplement 3). This analysis demonstrates that the effect on nurf-1 expression is subtle, as we were unable to identify significant differences in expression at timepoints that when phenotypic effects were observed (i.e.

spermatogenesis and expression of other genes). While disappointing from a mechanistic perspective, this serves as an interesting demonstration that the effect of a genetic variant on fitness can be significant despite having a subtle effect on transcription.

2) The authors prove that the full-length transcript of nurf-1 is not needed, and they provide evidence that one clade of Caenorhabditis have split the ancestral, single complex nurf-1 gene. However, there remains the question of whether nurf-1's 5'-ward and 3'-ward halves remain connected as part of one operon, with SL2 splicing to the 3'-ward half. Moreover, why the splitting of nurf-1 happened is a) not clear, and b) not connected to the domestication phenotype. It is thus a curiosity unrelated to a major biological event (thus far).

We do not believe that this is the case for the following reasons: 1. In *C. elegans*, the b and d isoforms are expressed from independent promoters. For these genes to be expressed as an operon, additional genetic changes must have fixed. 2. Previously published work from the Ellis lab supports that sl1 leader sequence is spliced to nurf-1-2 (Chen et al. – “Dependence of the sperm/oocyte decision on the nucleosome remodeling factor complex was acquired during recent *Caenorhabditis briggsae* evolution”). 3. We searched RNAseq data for sl2 sequence in clipped reads directly upstream of nurf1-2 without success. In *C. briggsae* and *C. brenneri*, we were able to identify three reads (2 and 1, respectively) that matched sl1 sequence. 4. In *C. tropicalis*, nurf-1-1 and nurf-1-2 are separated by ~10kb (Figure 6—figure supplement 3), which would be quite unusual for genes expressed in a operon. 5. In some species, nurf-1-1 and nurf-1-2 display different levels of expression, consistent with independent promoters being responsible for their expression (e.g. *C. brenneri* in Figure 6—figure supplement 3).

Finally, we note that even if the two genes are expressed in a single operon, this would not take away from the main parts of the story as they are free to evolve independently in the shared exon region.

3) Although some evidence of accelerated evolution of the overlap region between Yin and Yang isoforms is provided, there is no compelling evidence that there was an "adaptive conflict" that needed to be resolved and was indeed resolved. The null hypothesis of "both work, and developmental systems drift happens" cannot be rejected. This issue is at the core of your article's claim to biological importance.

First, we want to clarify with the reviewers (in case it was not clear from the 1^st^ submission) that there are two separate arguments we are making as to why this the duplication might be beneficial. 1. Loss of non-functional (a and q) transcripts necessary for expressing both functional isoforms (b and d) using the shared exons. 2) Adaptive conflict in the shared exon region.

While we agree with reviewers that we cannot reject developmental drift and must rely on circumstantial evidence to make the case, we also note that this is almost always the case for any analysis of extant species. Ideally, we would have biochemical information about activity changes in the shared exon regions, however, this is out of scope for this paper. Besides providing evidence that the duplicated regions experience accelerated evolution, we believe that including the evidence that nurf-1 is targeted by laboratory evolution strengthens the case that variation in nurf-1 is under selection in natural environments as well. Additional evidence that nurf-1 is under selection in the wild was previously provided by the Ellis lab’s paper (Chen et al. – “Dependence of the sperm/oocyte decision on the nucleosome remodeling factor complex was acquired during recent *Caenorhabditis briggsae* evolution”).

We have rewritten the Discussion to make all of this clearer.

All three reviewers agreed that the paper has abundant experimental data of high quality. The problem is that these data are not convincingly linked to the explanations claimed, and the results seem to fall naturally into two distinct papers (one on domestication alleles, the other on the molecular biology and evolution of isoforms arising from the genetically complex nurf-1 locus). Unless the three points above can be addressed, this work may fare better as two shorter studies focused on these two topics.We have attached the individual reviews by the three reviewers below.Reviewer #1:Using molecular biology, reverse genetics, and transcriptomics, Xu et al. make a compelling case for the evolution of two isoforms of nurf-1/BPTF in Caenorhabditis elegans that subdivide functions within this gene; they go on to show that a subgenus of Caenorhabditis species have evolved two entirely separate genes from these isoforms. Their observations provide a coherent explanation for many functions of the nurf-1 gene that had been intermittently seen over the years, but not reconciled. Their analysis of nurf-1 provides a beautiful instance of evolutionary adaptation, both in the sense of nurf-1's evolving into two functional isoforms, and in the sense that nurf-1 itself (as the authors show) mediates phenotypic tradeoffs in life history between early spermatocyte production and later oocyte production, which ultimately affects speed and output of the C. elegans reproductive life cycle that have different optima in different laboratory environments (plate culture for the N2 strain, versus liquid culture for the LSJ2 strain).The experimental work described here is thorough, rigorous, and adroit.The manuscript is very well composed. Background material is scholarly but appropriately concise. Generally, text is clean and direct. I was struck by the clarity with which the figures presented data visually. Papers such as this one that rely on genomic or transcriptomic analysis can be, in my experience, quite obtusely written or illustrated with figures that do not convey data well; Xu et al. have opposed these tendencies, for which I thank them.I was impressed with the authors' efficient use of previously published RNA-seq data for heat shock expression, and of prepublication Nanopore long-RNA-seq data from Roach et al., 2019. By the time Xu et al. is published, I hope that the preprint Roach et al. can be cited as a published journal article.In the text, I encountered only two points that I found difficult to decipher, a few missing data points, and one issue about nomenclature.The first unclear point involved strain PTM417 versus strain PTM88. In Figure 1D, one of the strains shown is PTM417, but this is referred to as PTM88 in the figure legend. Reading the text carefully twice and reviewing the Materials and methods did not clarify for me which was the correct strain for Figure 1, although I suspect the correct strain is PTM417 rather than PTM88. In particular, the description of PTM417 in the Strains list (Materials and methods) is not consistent with the description of how PTM417 was constructed (subsection “CRISPR-generated allelic replacement lines (ARLs)”). I request that the authors entirely resolve this confusion.

The reviewer correctly points out the errors in the figure legend. PTM417 is a derivative of PTM88, constructed by backcrossing out the spe-9 mutation while leaving the 60bp deletion.

The second unclear point involved the "Normalized Size" assay shown in Figure 3C. Two readings of the manuscript and one careful reading of the Materials and methods did not clarify, for me, what was meant by 'size' (though my guess was that it was body size). I went back to their key reference for many methods (Large et al., 2016) and concluded that the authors actually mean "Normalized body size". This should be made completely clear to the reader, both in the y-axis labeling of Figure 3C and in the legend text for Figure 3C, as well as in the Materials and methods.

A figure legend for 3C was accidentally deleted from the submitted manuscript. Additionally, the Materials and methods section was poorly written and incorrect. The label and legend for Figure 3C and Materials and methods section have been fixed.

One missing data point is in Table 2. This table should list a stop-codon location for kah11 in the isoform NURF-1.F, and only in that one isoform. Instead, Table 2 currently gives no stop-codon mutations for kah11 at all. In addition, Figure 3D (which is supposed to show all stop codons in their structural context) also omits kah11. These omissions from both Table 2 and Figure 3D should be corrected.

Corrected.

A second missing data point appears to be in the genotype table of Figure 4B. For the genotype +/kah96, this table claims that there are 0 functional copies of the isoform nurf-1.d; however, I do not see how this can be possible, given that the wild-type allele [+] should encode at least 1 functional copy of nurf-1.d. Unless there is something that I am badly misunderstanding (always possible), please correct this.

Corrected.

About nomenclature: the authors have dubbed the NURF-1.B isoform "Yin" and the NURF-1.D isoform "Yang". I assume that this is because of their order of expression (NURF-1.B is expressed before NURF-1.D), so that calling them "Yin and Yang" follows their times of activity. However, I found the nomenclature confusing because Yang is associated with stereotypically masculine traits and entities, whereas Yin is associated with feminine ones. Yet, the authors' current nomenclature has Yin assigned to the isoform promoting spermatogenesis (NURF-1.B), while Yang is assigned to the isoform promoting oogenesis (NURF-1.D). Since I assume that the authors would like to make it *easy* for people to properly remember which isoform does what, I would strongly encourage them to switch their nomenclature (so that NURF-1.B gets called Yang, and NURF-1.D gets called Yin).

We agree with the reviewer and have modified the text accordingly.

Reviewer #2:Here, Xu et al. report several splice isoforms of the C. elegans nurf-1 gene and show that they have different, possibly opposite, effects on certain aspects of gametogenesis. This has implications for fitness. The authors also found that the complex nurf-1 locus has undergone a partial duplication in several close relatives of C. elegans, thereby segregating distinct functions of nurf-1 into two separate genes.1) The finding that gene expression profiles of N2 and ARL strains differ considerably at 52h, but apparently not at 60h is (Figure 1F) seems quite interesting. Surprisingly, it was not explored further. At least it should be commented on in a more elaborate way.

We have added an additional sentence in the Results (subsection “An N2-derived variant in the second intron of nurf-1 increases fitness and brood size in laboratory conditions”, last paragraph).

2) In the second paragraph of the subsection “The B and D isoforms have opposite effects on cell fate during gametogenesis”, the authors state that the observed reduction in the number of sperm is due to earlier sperm-to-oocyte switch. This is plausible, but other causes are also possible and I would encourage the authors to provide direct experimental support for this claim.

These experiments are rather laborious (DAPI staining and counting hundreds of sperm in ~50-60 animals). Since this point is not central to the paper, and in order to spend time addressing the main points brought up by reviewers above, we have not performed these experiments. We agree that other causes are possible and have added an additional sentence in the Results.

3) Are defects in sperm production of certain alleles of nurf-1 alleles (eg kah106) unique to hermaphrodites or are they seen in males as well? Conversely, are oocyte production defects of kah93 an issue in mutants that only produce oocytes? Addressing these questions could help to support the notion that different isoforms contribute to different aspects of the tradeoff between sperm and oocyte production. Are the observed defects in the numbers of gametes due to erroneous timing of the switch or some other problem? Can anything be said about the functions of different isoforms in gonochoristic species in which sperm vs. egg conflict is not a concern?

This is a very interesting question that we can address using a recently accepted paper in Genetics (https://www.genetics.org/content/early/2019/08/08/genetics.119.302462) which demonstrates a role for nurf-1 in determining sperm size. Using the n4295 allele (which removes much of the d isoform), we can demonstrate a role for the d isoform in playing a role in spermatogenesis (i.e. sperm size) in both males and hermaphrodites. However, the n4295 has an opposite effect on body-length in hermaphrodites (shorter) vs. males (longer). This indicates that nurf-1 has sex-specific roles, consistent with our central hypothesis that nurf-1 is a life history regulator and the fact that life history tradeoffs are influenced by sex. We would predict that the traits that nurf-1 controls will be sex and species specific (of course some will be shared). This is an area that we are interested in following up on.

4) The concluding sentence of the Results section claims that the rate of amino acid replacements has accelerated in the duplicated exons. I did not find a formal test supporting this assertion.

We have performed a test and have added this to the text.

5) Are the authors aware of this paper – Hughes, 1994?

We were not aware of this paper. We thank the reviewer for pointing this out and we have added this reference to the Introduction and Discussion.

Reviewer #3:In this paper, Xu et al. present a meticulous examination of the nurf-1 locus of C. elegans. The study presents several interesting findings:1) New evidence (beyond published studies) is presented indicating that nurf-1 has been a target of selection during domestication of laboratory strains, including the N2 and LSJ2 strains. In particular, the authors make a strong case for a major role of an intronic single nucleotide variant (SNV) in mediating adaptation to the NGM plate culture in the N2 lineage.2) The authors then shift to a detailed characterization of the various nurf-1 transcripts and their necessity for growth and sustained fertility. This is most impressively supported by a battery of engineered stop codons. The results strongly suggest that the nurf-1.b and nurf-1.d transcripts, which overlap slightly, are the key effectors of nurf-1 function, and that the full-length nurf-1.a transcript is likely dispensable.3) While isoform-restricted stop codons that reduce or eliminate nurf-1.b and nurf-1.d function both reduce self-fertility, they do so via opposite effects. nurf-1.b appears to be necessary to support robust spermatogenesis (i.e. it's loss leads to a partial Fog), while nurf-1.d has a role in promoting the switch from spermatogenesis to oogenesis (i.e. it is a partial Mog).4) In the clade of Caenorhabiditis species that includes C. brenneri, the partially overlapping transcripts have been completely separated via a lineage-specific duplication of the exons that were historically shared.Overall, this paper is a genetics tour de force. I do have a few suggestions, that if addressed, would tighten up the story:1) While there appears to be a surprisingly major effect of the intronic SNV on both global gene expression and fitness, nothing is said about how this change in a homopolymeric run alters nurf-1's own gene expression. One might expect this to impact nurf-1.b transcription or splicing, and this could, in turn, alter NURF-1.B levels. Can the authors provide any data about the functional impact of the SNV on nurf-1? For example, an isoform frequency chart like that for Figure 2C, but comparing PTM228 and the ARL(intron, LSJ2>N2), would be very informative as to the mechanism by which the intron SNV impact phenotype.

As discussed above, we have added this analysis to the paper.

2) We are told that "brood size of C. elegans hermaphrodites is an important trait for evolutionary fitness in laboratory conditions." Indeeed it is, but the timing also matters, with early progeny much more valuable than late progeny. Looking at the reproductive schedules in Figure 1—figure supplement 1 (and also in the Large et al., 2016), there is a real shift in timing. Can the authors model, or at least speculate on, the expected impact of this?

This was sloppy on our part. We agree that both timing and brood size are a life-history tradeoff and both matter for fitness. Previous modeling (Cutter, 2004) and experimental work (Hodgkin, 1991) has suggested that N2 brood size balances this trade off. We make this clearer in the text.

3) In the species that have a fully separated nurf-1.1 and nurf-1.2 genes, there is very little space between the two. Have the authors looked to see if the 5' end of the transcript from the downstream gene is spliced to SL2? If so, that would indicate these have formed an operon.

As discussed above, we do not believe that these genes are expressed as an operon, although we are unable to conclusively say they are not.